# TIER: Text-Image Entropy Regularization for CLIP-style models

**Anil Palepu**                                                    *apalepu@mit.edu*
*Harvard-MIT Health Sciences & Technology*

**Andrew L. Beam**                                        *andrew_beam@hms.harvard.edu*
*Harvard University*

## Abstract

In this paper, we study the effect of a novel regularization scheme on contrastive language-image pre-trained (CLIP) models. Our approach is based on the observation that, in many domains, text tokens should only describe a small number of image regions and, likewise, each image region should correspond to only a few text tokens. In CLIP-style models, this implies that text-token embeddings should have high similarity to only a small number of image-patch embeddings for a given image-text pair. We formalize this observation using a novel regularization scheme that penalizes the entropy of the text-token to image-patch similarity scores. We qualitatively and quantitatively demonstrate that the proposed regularization scheme shrinks most of the pairwise text-token and image-patch similarity scores towards zero, thus achieving the desired effect. We demonstrate the promise of our approach in an important medical context, chest x-rays, where this underlying sparsity hypothesis naturally arises. Using our proposed approach, we achieve state of the art (SOTA) average zero-shot performance on the CheXpert and Padchest chest x-ray datasets, outperforming an unregularized version of the model and several recently published self-supervised models.

## 1    Introduction

Self-supervised vision models that leverage paired text data such as the contrastive language-image pre-trained (CLIP) model (Radford et al., 2021; Zhang et al., 2020) have demonstrated very impressive zero-shot classification performance in a variety of domains (Radford et al., 2021; Tiu et al., 2022; Boecking et al., 2022; Palepu & Beam, 2022). Specifically, users can leverage the unified text and image embedding space for zero-shot classification by providing relevant text queries and assessing image embedding similarities (Radford et al., 2021; Tiu et al., 2022; Kumar et al., 2022).

The CLIP architecture consists of a vision encoder, typically a CNN (He et al., 2016) or vision transformer (Dosovitskiy et al., 2020), and a text encoder, typically a text transformer (Vaswani et al., 2017). Each encoder produces a single embedding in the joint embedding space that aims to summarize all of the relevant information in their respective modality. A recent CLIP-style architecture from Boecking et al. (2022), which was built on chest x-ray (CXR) data, allows for a more fine-grained representation of images by projecting the final ResNet block's output to the joint embedding space prior to doing a global average pooling. As a result, this model produces a set of local or *patch* embeddings which can be indicative of not just *if* a text and image align, but also *roughly where* they align. As an example, in a CXR positive for cardiomegaly (an enlarged heart), the patch embeddings near the heart would likely have a higher cosine similarity to the text embedding of "an enlarged heart" than other regions would.

In certain domains such as CXRs, it is clear that important visual features tend to be confined to a relatively small portion of the image. For example, cardiomegaly is primarily identified in the lower left portion of the chest, but a CXR captures many anatomical regions beyond this area. At the same time, complex image captions could describe multiple diverse clinical findings which are unlikely to all correspond to the exact same

CXR regions. In this work, we propose a method to encode these observations into any CLIP-style model that can produce individual image-patch embeddings and text-token embeddings. To do so, we introduce text-image entropy regularization (TIER), which encourages text-token embeddings and image-patch embeddings to be less 'promiscuous' by regularizing the entropy of a softmaxed distribution of similarity scores. This regularization can be modulated by adjusting two hyperparameters, and because it is based on entropy, it is robust to positional shifts in both the text and the images.

We implement our TIER method leveraging the pre-trained architecture from Boecking et al. (2022), and demonstrate both qualitatively and quantitatively that our regularization method shrinks the text-token and image-patch similarity scores towards zero. We evaluate the resulting model by comparing it to an equivalent unregularized baseline, a fully-supervised baseline, and several state-of-the-art, CLIP-style CXR benchmarks (Tiu et al., 2022; Wang et al., 2022b). We demonstrate that our method results in zero-shot accuracy improvement across a wide range of clinical findings, setting a new state of the art in many instances.

In summary, we make the following contributions:

- A novel regularization scheme applicable to any CLIP-style model that produces local image and text embeddings. The regularization term shrinks the text-token and image-patch similarity scores to encourage sparser image-text similarities.

- We establish a new state of the art (SOTA) zero-shot classification AUC on the CheXpert test set, surpassing recently introduced self-supervised models and several previously published fully supervised ones.

- We also establish a new SOTA for average zero-shot classification AUC on the Padchest dataset, which measures classification performance across many diverse clinical findings.

## 2 Related Works

Many groups have made efforts to promote more fine-grained alignment of images and text in CLIP-style models (Yao et al., 2021; Li et al., 2022a; Zhong et al., 2022; Wang et al., 2022a; Huang et al., 2021; Li et al., 2022b). Several of these approaches require a separate region proposal/object detection network (Zhong et al., 2022; Li et al., 2022b;a); while typically effective for natural images, these objection detection models have not been equivalently validated in medical domains like CXRs.

Other approaches (Wang et al., 2022a; Huang et al., 2021) aim to modify the contrastive loss in order to better align local representations, but unlike our approach, do not directly aim to induce *sparsity*. For example, while Huang et al. (2021) and Wang et al. (2022a) both include a local contrastive loss, their approaches potentially allow tokens to be similar to all of the cross-modal tokens. Furthermore, Huang et al. (2021) was shown to have poor zero-shot performance when evaluated by Tiu et al. (2022), while Wang et al. (2022a) is not yet publicly available for evaluation.

Unlike the previously described approaches, and like our TIER method, the approach in Yao et al. (2021) does induce sparsity at the token and patch level. However, their approach more aggressively forces sparsity by only considering the maximum similarity text token for each image token and vice versa. Conversely, our approach allows us to flexibly modulate the level of sparsity using two tune-able hyperparameters (which could allow us to mimic the effect of Yao et al. (2021) if set extremely high).

## 3 Methods

### 3.1 Data

We utilized the MIMIC-CXR-JPG (Johnson et al., 2019) dataset to train our models and the CheXpert (Irvin et al., 2019) and Padchest (Bustos et al., 2020) datasets to evaluate them.

The MIMIC-CXR dataset (Johnson et al., 2019) consists of 377,095 CXR samples from 65,379 different patients. Many patients have multiple radiological studies within the dataset, with a single study often

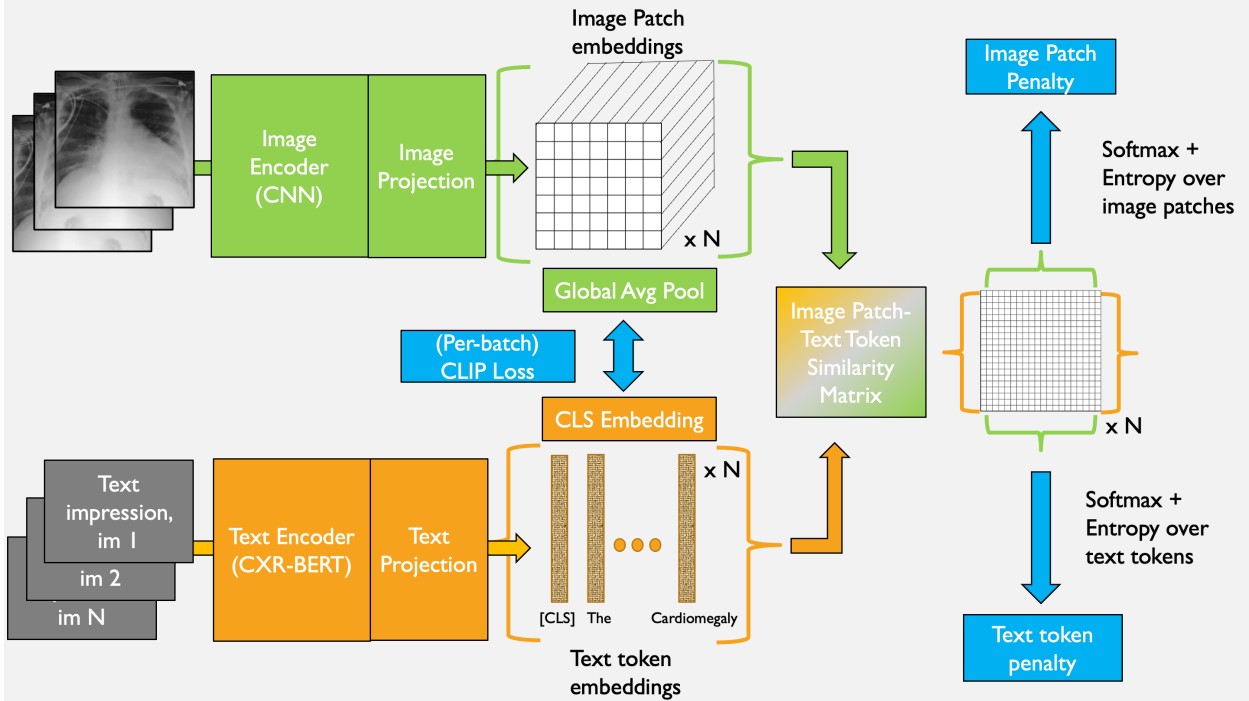

Figure 1: An overview of TIER, our regularized training method.

containing both a frontal and lateral CXR view. These CXRs were evaluated by radiologists, who wrote detailed reports on the clinical findings they observed as well as a sentence or two describing their overall impression of the imaging. We extracted these *impression* sections from the radiology reports to use as the paired text for our image input. We dropped any samples that were missing this impression section, leaving us with a total of 319,446 CXR-impression pairs. We split these MIMIC-CXR image-text pairs into training and validation subsets (with approximately 90% training data) and ensured that no patient was represented in both subsets.

For evaluation, we utilized the separate CheXpert (Irvin et al., 2019) dataset with pre-defined validation and test splits which consisted of 234 and 668 CXRs respectively. These subsets of CheXpert have 14 different clinical labels, determined by consensus of 3 and 5 radiologists respectively. We benchmarked our models' thresholded predictions using labels from an additional 3 radiologists available in the CheXpert test set. For the purposes of our evaluation, we only considered the following 5 clinical labels: Cardiomegaly, Edema, Consolidation, Atelectasis, and Pleural Effusion. These labels were the five competition tasks from the CheXpert competition and the most commonly attempted tasks in the literature, making them a natural set for comparison. We also extracted these labels from the MIMIC-CXR dataset, but we only used them when training our fully supervised CNN baseline; our contrastive models did not have any access to these labels.

We additionally evaluated our models with the Padchest dataset (Bustos et al., 2020), of which we only considered the subset of 39,053 CXRs that were labeled by radiologists. There were over a hundred different labels present in these CXRs, but we focused on the set of 57 labels that were present with frequency of at least 50 in our selected subset, as was done by Tiu et al. (2022).

All images were resized to $224 \times 224$ pixels with 3 RGB channels. At train time, we performed random data augmentations including random resizing, cropping, affine transformation, and color jitter, while at test time, we simply resized images to $256 \times 256$ before center cropping to $224 \times 224$.

## 3.2 Model Architecture

We based our model on the BioViL architecture (Boecking et al., 2022), which consists of a pre-trained ResNet-50 architecture as the vision encoder and "CXR-BERT-specialized", a transformer, as the text encoder. This model differed from the original CLIP architecture in that it consisted of a radiology-specific text encoder (CXR-BERT-specialized) and was trained with an additional MLM loss, among several other changes (Boecking et al., 2022). This model was also trained using MIMIC-CXR and importantly did not have access to the CheXpert or Padchest datasets, which we used for evaluation.

For our purposes, the most critical feature of the BioViL model is that the final ResNet-50 block provides embeddings that correspond to local, connected regions of the input image (see the green path on the top of Figure 1). Thus, in addition to the single global image embedding, for an input image of size $224 \times 224$ this model also produces a set of 49 embeddings in a $7 \times 7$ grid, which all share the same joint feature space as the global image embedding. The number of embeddings is a function of the original input size (a larger image input would yield more embeddings) as well as the choice to use the final ResNet block output (using an earlier output would lead to more fine-grained local embeddings). A single multi-layer perceptron with one hidden layer was used to project each local embedding to the joint feature space. We call the output of the ResNet block the *patch* embeddings as they correspond to regional patches of the image.

The text transformer naturally produces a text token embedding for each input token to the model. We use a single multi-layer perceptron with one hidden layer to project these text token embeddings to the joint feature space. The projected embedding from the first text token, [CLS], is contrasted with the global image embedding as is done with typical contrastive language-image pre-trained models (Radford et al., 2021; Palepu & Beam, 2022; Zhang et al., 2020). For a training batch of image-text pairs $(x^I, x_T)$, we use the standard CLIP loss as described in Radford et al. (2021). We add additional penalty terms, described in the following section, to regularize our model beyond this standard CLIP loss. The pseudocode for our method is described in the appendix Fig. 6

## 3.3 TIER: Text-Image Entropy Regularization of Image-Patch and Text-Token Similarity Scores

TIER works by first computing a matrix of image-patch and text-token similarities. Specifically, consider an example image-text pair that has $I = \{I_1, \ldots, I_P\}$ image-patch embeddings (in our case, $P = 49$), and has $T = \{T_1, \ldots, T_T\}$ text-token embeddings ($T$ can vary for each sample as captions can be different lengths). We compute the image patch-text token similarity matrix $S$ by computing a $T \times P$ matrix of cosine similarities between each image-patch embedding and text-token embedding. The embeddings for each input modality are the outputs of an encoder model that is specific to that input, e.g. a CNN or vision transformer for images and a BERT-style transformer for text. Importantly, we select encoders that provide embeddings at the token level, i.e., image-patch embeddings and text-token embeddings. Row $i$ of $S$ indicates the cosine similarity between a text-token $T_i$ and each image patch in $I$. The columns of $S$ likewise indicate the cosine similarity between a given image patch $I_j$ and each text token in $T$.

Recall, the goal of our approach is to shrink the elements of $S$ such that each text token is similar to a relatively small number of image patches. To do this, we introduced an entropy-based penalty term that induces shrinkage on the elements of $S$. First, we perform a row-wise softmax of $S$ and measure the entropy between a text token $T_i$ and all of the image patches in $I$, shown below:

$$\mathcal{H}(T_i, I) = \sum_{j=1}^{|P|} -p_j * \log(p_j) \tag{1}$$

where $p_j$ is the probability produced by the softmax of the row of $S$ corresponding to $T_i$. This term will be maximized when each $p_j$ is $\frac{1}{P}$, implying that all of the image patch embeddings have equal similarity to $T_i$.

Next, we apply the same procedure to the columns of $S$, applying a column-wise softmax over the text-token similarities to produce probabilities $p_1$ to $p_T$ for each image patch $I_i$ and calculating the entropy of these probabilities as follows:

$$\mathcal{H}(T, I_j) = \sum_{i=1}^{|T|} -p_i * \log(p_i) \tag{2}$$

We average the $N \times T$ image-patch entropies $\mathcal{H}(T_i, I)$ and the $N \times P$ text-token entropies $\mathcal{H}(T, I_j)$ to produce an *image-patch penalty* and *text-token penalty* for the batch. We control the effects of these penalties on training by weighting them with hyperparameters $\lambda_t$ and $\lambda_p$ respectively, adding the weighted penalties to the CLIP loss to compute the total loss.

A grid search over the range $[0, 0.25]$ was used to set the hyperparameters ($\lambda_p = 0.2$, $\lambda_t = 0.1$) for our regularized model. Specifically, we trained our contrastive models for just a single epoch on MIMIC-CXR with pairs of $\lambda_p$ and $\lambda_t$, and chose the pair that maximized zero-shot AUC on the validation set. These results are available in the appendix Appendix A.3. Both the training procedure and zero-shot classification method are described in later sections.

### 3.4    Training Details

We begin with the pretrained BioViL architecture and model weights, "CXR-BERT-specialized" (Boecking et al., 2022), which has already been trained with contrastive learning on the MIMIC-CXR dataset. In this original training, only frontal images were used, and they used a masked language model (MLM) loss in addition to the CLIP loss. Starting with this pretrained model, we train two separate CLIP-style models: A regularized model in which $\lambda_p = 0.2$ and $\lambda_t = 0.1$, as well as an unregularized baseline model, in which $\lambda_p = \lambda_t = 0$. Despite only minor changes (further training on MIMIC-CXR, inclusion of lateral CXRs, omission of the MLM loss, freezing of early text encoder layers), our unregularized baseline significantly outperformed the publicly available pretrained model from Boecking et al. (2022) as seen in Appendix A.6.

All aspects of model training are identical between our regularized and unregularized models, other than the additional penalty terms. For both models, we freeze the first 8 layers of the BERT encoder, while leaving the rest of the text encoder and vision encoder unfrozen. Each model is trained for 30 epochs using the loss described in the previous section with a learning rate of 0.0001 and batch size of 32.

We also train a fully supervised CNN baseline, which utilizes the same vision encoder as the contrastive models but has a multilayer perceptron with one hidden layer and five outputs. This supervised baseline still uses MIMIC-CXR for training, but instead of text, it is trained with labels using binary-cross entropy loss with a learning rate of 0.0001 and batch size of 32.

### 3.5    Zero-shot classification

We employ a zero-shot classification procedure that leverages our text and image encoders to identify labels of interest in the images. Our method begins with the user selecting $K_p$ positive and $K_n$ negative queries for the label of interest, $Q$. Positive queries $\{Q_{p1}, ..., Q_{pK_p}\}$ are text descriptions indicative of the presence of that label, while negative queries $\{Q_{n1}, ..., Q_{nK_n}\}$ are text descriptions indicative of the absence of that label; examples which we used for the five CheXpert labels are detailed in Tab. 8 in the appendix. We pass each positive query through the text encoder, project their [CLS] token embeddings to the joint embedding space, and then average these projected embeddings and re-normalize to a unit norm. We do the same for the negative queries so that we have a single positive $\overline{Q_p}$ and negative $\overline{Q_n}$ query embedding associated with each label that we wish to classify:

$$\overline{Q_p} = \frac{\sum_{j=1}^{K_p} Q_{pj}/K_p}{||\sum_{j=1}^{K_p} Q_{pj}/K_p||} \qquad\qquad \overline{Q_n} = \frac{\sum_{j=1}^{K_n} Q_{nj}/K_n}{||\sum_{i=1}^{K_n} Q_{nj}/K_n||} \tag{3}$$

For any input image we wish to classify, we use the image encoder to compute its projected global image embedding $E_{img}$ (normalized to unit norm) and take the dot product of this global image embedding with both the positive and negative query embeddings $\overline{Q_p}$ and $\overline{Q_n}$ for every label we wish to predict. We subtract

these positive and negative cosine similarity scores to get a zero-shot classification score, $Z_Q$, for our label of interest.

$$Z_Q = (E_{img} \cdot \overline{Q_p}) - (E_{img} \cdot \overline{Q_n}) \tag{4}$$

Importantly, our zero-shot classification output cannot be interpreted as a probability as its range is actually between $[-2, 2]$. We are primarily interested in assessing discriminative performance of our zero-shot classifiers, so interpretation as a probability is not necessary; however, if one desired a probability output, they could simply apply a softmax to the positive and negative similarity scores as was done by Tiu et al. (2022) instead of subtracting these scores.

## 4 Results

### 4.1 Visualization of the effect of regularization

Qualitatively, our regularization method is able to achieve the desired shrinkage between image patches and text tokens. Fig. 2 and Fig. 3 show patch-level zero-shot classification scores (i.e., the score between each image patch and the global [CLS] text token) overlaid on top of two CXRs, one with cardiomegaly and one without. In these heatmaps, red is indicative of a higher zero-shot score, blue is a lower score, while gray is a more neutral score.

Important differences between the regularized and unregularized models are apparent when we examine the distribution of blue and red regions of the heatmaps in Fig. 2 and Fig. 3. For the cardiomegaly-positive image (Fig. 2), the regularized model has high score primarily on the lower left side of the patient's chest (which corresponds to lower right side of the image), where their heart is located. Likewise, the regularized model shows low similarity (blue) on the other of their lower chest where one could expect to see some changes in more extreme cases of cardiomegaly. These similarity scores seem rational and are clinically justifiable. On the other hand, while the unregularized model also displays some signal in the clinically relevant regions, it has significantly more extreme similarity scores scattered throughout the image well beyond the heart-adjacent regions. We see similar results on an example for pleural effusion detection (Fig. 3), as well as in additional examples presented in the appendix (Fig. 7, Fig. 8, Fig. 9).

### 4.2 Distribution of image-patch similarity scores to global [CLS] text token

To further explore the effect of our regularization method on the image-patch similarities, we utilize a set of 160 positive image-text pairs from MIMIC-CXR. In Fig. 4, we plot the similarity of the projected [CLS] token embeddings to the 49 image-patch embeddings from the corresponding image. In this figure, the image-patch similarities were ranked in descending order before being plotted, with error bars indicating the standard deviation across the 160 samples. We can see that the regularized model on average has significantly lower similarities to the patch embeddings than the unregularized model. To better visualize these differences, we produced Fig. 5, which displays the same information as Fig. 4 except each patch similarity was normalized by dividing the similarity by the sum of all patch similarities in the entire image. In this plot, we can clearly see that the regularized model tends to have a few patches with relatively higher similarities to the [CLS] token embedding, and many with relatively lower similarities; this supports our hypothesis that our regularization scheme shrinks token-level similarity in the model, thus achieving a lower entropy.

### 4.3 Zero-shot classification

Next, we evaluated our zero-shot classification method for both the regularized and unregularized models on the held-out CheXpert test set. Our primary benchmark for these models is the 'CheXzero' model (Tiu et al., 2022), which recently achieved SOTA zero-shot AUC on this task. We use weights from the checkpoint that achieved the highest AUC on the CheXpert validation set. We also evaluate another recent self-supervised model, MedCLIP (Wang et al., 2022b), with the caveat that this model is not strictly zero-shot because the authors utilized clinical labels during their training process. Additionally, we evaluate a fully supervised CNN that uses our vision encoder with an additional classification head. We bootstrap 1000 times, randomly

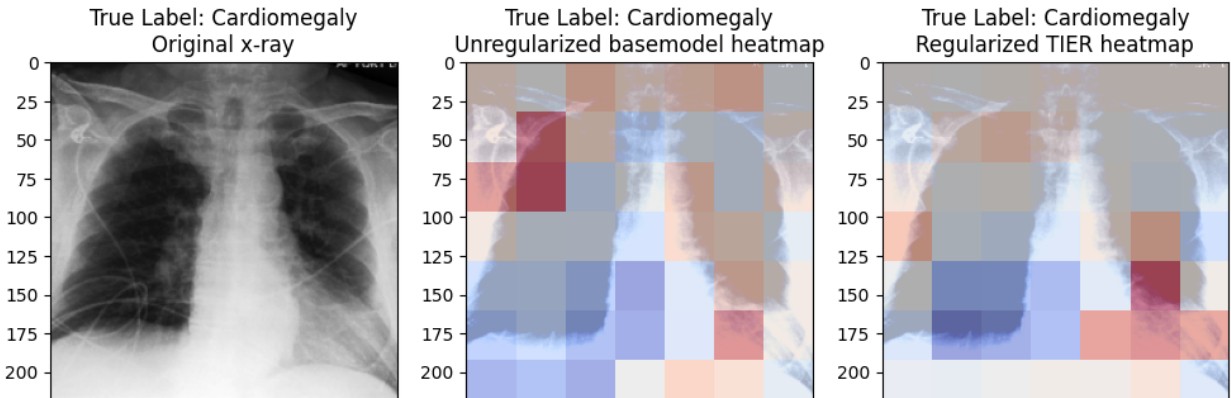

Figure 2: **The penalty term induces shrinkage in the image patch-text token similarity scores**. A CXR positive for cardiomegaly, overlaid with heatmaps displaying zero-shot cardiomegaly score for the unregularized (center) and regularized (right) models. Gray corresponds to a neutral (close to zero) zero-shot score, while red is a higher score and blue is a lower score. As can be seen by comparing the middle and right figures, the regularized model focuses more on the relevant regions and less on the periphery. Note: In this instance cardiomegaly is located in the lower right portion of the image.

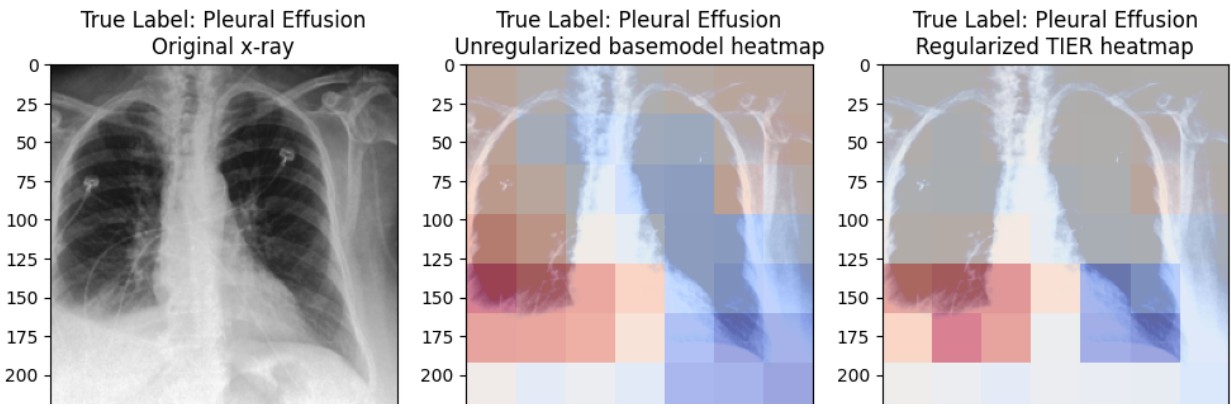

Figure 3: A CXR with pleural effusion, overlayed with heatmaps displaying zero-shot pleural effusion score for the unregularized (center) and regularized (right) models.

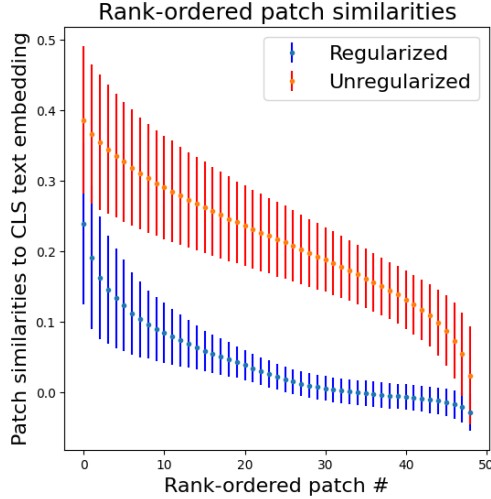

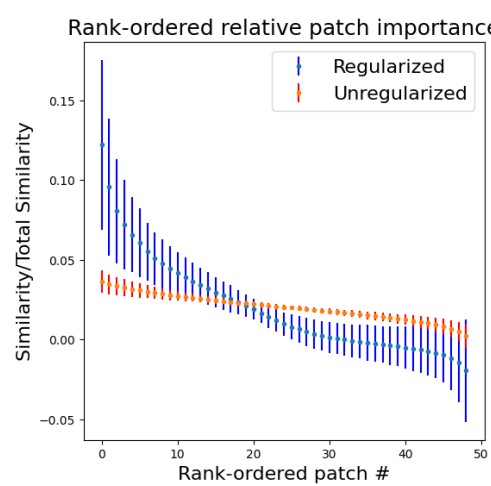

Figure 4: The similarities of each patch to the CLS embedding for a set of 160 MIMIC-CXR images, sorted in descending order.

Figure 5: The same plot as Fig. 4 in which each similarity is divided by the sum of the total similarity across all patches for that image.

| Label | Regularized TIER | Unregularized Basemodel | CheXzero | MedCLIP | Fully Supervised CNN |
|---|---|---|---|---|---|
| Average | **0.903336** | 0.89721 | 0.89349 | 0.87708 | 0.88877 |
| Cardiomegaly | **0.91714** | 0.89239 | 0.88340 | 0.83911 | 0.86400 |
| Edema | **0.92423** | 0.90729 | 0.89424 | 0.91242 | 0.92236 |
| Consolidation | 0.89712 | 0.91213 | **0.91318** | 0.88653 | 0.86003 |
| Atelectasis | **0.86531** | 0.85741 | 0.84299 | 0.79423 | 0.85869 |
| Pleural Effusion | 0.91290 | 0.91681 | **0.93368** | 0.95313 | 0.93876 |

Table 1: Average AUCs for various models on 1000 bootstraps of CheXpert Test. The highest zero-shot AUC is bolded (the three on the left are performing zero-shot classification, in which they have never previously seen any labels in the training set). As seen in the appendix (Tab. 4, Tab. 5, and Tab. 6), all differences between zero-shot models are statistically significant except for Unregularized vs CheXzero for Consolidation. Though MedCLIP is trained with contrastive learning, it also utilizes labels during training so we do not consider it to be fully zero-shot.

sampling the test set with replacement and evaluating the mean AUC performance of each model over these 1000 bootstraps. These results can be seen in Tab. 1, which demonstrates that our regularized model achieves SOTA zero-shot AUC; regularization offers a modest bump in average AUC performance. Excluding "Unregularized vs CheXzero for Consolidation", all pairwise AUC differences between the zero-shot models are statistically significant according to a two sample t-test for the difference of means. These zero-shot models are also competitive against three reference radiologists according to their Matthews's correlation coefficient (MCC) and F1 scores, as seen in the appendix (Fig. 10).

We use the Padchest dataset to evaluate a broader set of findings, specifically looking at the 57 findings with $n \geq 50$ from the radiologist-labeled subset of Padchest. We constructed positive label queries using the phrase "X is present.", while we constructed negative label queries with the phrase "No X.", replacing X with the label of interest. The notable exception was when we classified "normal" images; in this instance, we used "Abnormal findings." as the negative query. Tab. 2 details the Padchest results for the regularized, unregularized, and CheXzero models. As seen in the appendix (Tab. 9, Tab. 10), our regularized TIER model achieves statistically significant boosts in performance on average as well as for the majority of Padchest findings when compared head-to-head with CheXzero and our unregularized baseline model.

### 4.4 Zero-shot COVID-19 diagnosis

We also evaluate our model COVID-19 detection, which is a diagnosis not present in any of our training data. As a result, our models cannot rely on the actual label itself (i.e., the word cardiomegaly in a text query), and therefore the diagnostic capability of our models on this task can be fully attributed to its ability to recognize the descriptive attributes being queried. Furthermore, discriminating COVID-19 and non-COVID-19 pneumonia from chest imaging is a non-trivial task, with one study reporting just a 74% average accuracy for three radiologists using chest CT for this task (Bai et al., 2020).

We created queries to discriminate COVID-19 and non-COVID-19 pneumonia based on differences mentioned in the literature (Bai et al., 2020; Borghesi & Maroldi, 2020). For the positive COVID-19 query, we used the query "Ground glass opacities and consolidation with peripheral distribution with fine reticular opacity and vascular thickening.", and for the negative COVID-19 query, we used "Pleural effusion present with lymphadenopathy and consolidation with central distribution." (which were described by Bai et al. (2020) as findings more specific to non-COVID-19 pneumonia). We achieved zero-shot AUCs of 0.759, 0.753, and 0.752 with the regularized, unregularized, and CheXzero models respectively on discriminating COVID-19 from non-COVID pneumonia within the COVID-QU-Ex Dataset (Tahir et al., 2021; 2022).

This performance indicates that we can leverage our model for difficult multi-class classification tasks by simply providing English descriptions of the class-discriminating features. Furthermore, this procedure can easily extend to other labels, meaning self-supervised vision-language architectures such as these could be leveraged to diagnose novel diseases if their presentation on imaging can be described.

| Label | Count | Regularized AUC | Unregularized AUC | CheXzero AUC |
|---|---|---|---|---|
| Average AUC | 39053 | **0.755420** | 0.742534 | 0.726306 |
| Number of Evaluations Won (Percent) | 57 | **20 (35.1%)** | 18 (31.6%) | 18 (31.6%) |
| endotracheal tube | 284 | 0.979606 | 0.956634 | **0.98295** |
| pleural effusion | 1748 | 0.942045 | 0.930292 | **0.950519** |
| pulmonary edema | 87 | 0.941289 | 0.945415 | **0.95646** |
| heart insufficiency | 546 | 0.926097 | **0.927177** | 0.917819 |
| pulmonary fibrosis | 166 | **0.951654** | 0.944147 | 0.921793 |
| cardiomegaly | 3746 | 0.883692 | 0.883339 | **0.890467** |
| vascular redistribution | 129 | **0.877236** | 0.872019 | 0.750592 |
| consolidation | 364 | **0.878342** | 0.849872 | 0.865175 |
| hilar congestion | 601 | **0.855435** | 0.850243 | 0.825707 |
| pulmonary mass | 247 | 0.844107 | **0.872299** | 0.842056 |
| cavitation | 122 | **0.857639** | 0.794295 | 0.853367 |
| alveolar pattern | 1353 | **0.87631** | 0.816974 | 0.763811 |
| calcified pleural thickening | 102 | **0.859651** | 0.84287 | 0.850707 |
| lung metastasis | 89 | **0.877375** | 0.860837 | 0.827675 |
| emphysema | 376 | 0.717841 | 0.718377 | **0.830578** |
| interstitial pattern | 1907 | 0.835144 | **0.840368** | 0.816432 |
| costophrenic angle blunting | 1683 | 0.769921 | **0.808131** | 0.69029 |
| COPD signs | 4823 | 0.650859 | 0.652912 | **0.751217** |
| tuberculosis | 59 | 0.838961 | **0.843741** | 0.7978 |
| atelectasis | 676 | 0.781707 | 0.791507 | **0.809232** |
| reticular interstitial pattern | 72 | 0.844479 | **0.867637** | 0.822429 |
| pneumonia | 1780 | **0.813796** | 0.796614 | 0.773941 |
| lobar atelectasis | 168 | 0.808411 | **0.815725** | 0.775147 |
| normal | 12694 | 0.776328 | **0.790588** | 0.753171 |
| pleural thickening | 213 | **0.784428** | 0.754608 | 0.752537 |
| reticulonodular interstitial pattern | 51 | **0.862346** | 0.838374 | 0.841414 |
| infiltrates | 1456 | 0.742854 | 0.735399 | **0.747836** |
| hypoexpansion | 166 | 0.853423 | **0.871452** | 0.794564 |
| hypoexpansion basal | 119 | **0.889652** | 0.874477 | 0.8018 |
| humeral fracture | 81 | 0.742305 | 0.672935 | **0.749084** |
| pneumothorax | 98 | 0.730643 | 0.728547 | **0.777442** |
| multiple nodules | 102 | 0.790815 | **0.852951** | 0.716911 |
| hyperinflated lung | 197 | 0.700879 | 0.667704 | **0.713202** |
| bronchiectasis | 667 | 0.734643 | **0.743998** | 0.690117 |
| adenopathy | 136 | 0.678726 | **0.73105** | 0.703924 |
| mediastinal enlargement | 106 | 0.72538 | 0.666796 | **0.759299** |
| laminar atelectasis | 1378 | 0.67343 | **0.687839** | 0.679276 |
| vertebral compression | 126 | 0.723955 | **0.734413** | 0.646448 |
| rib fracture | 140 | 0.689835 | 0.668069 | 0.691037 |
| tuberculosis sequelae | 185 | **0.796895** | 0.773832 | 0.584302 |
| hilar enlargement | 447 | **0.721779** | 0.714687 | 0.678564 |
| tracheal shift | 180 | 0.615827 | 0.500734 | **0.634359** |
| mediastinal mass | 74 | **0.709825** | 0.409473 | 0.647695 |
| central vascular redistribution | 63 | **0.728932** | 0.567387 | 0.354491 |
| vertebral fracture | 104 | 0.791375 | **0.86009** | 0.499654 |
| superior mediastinal enlargement | 153 | 0.551017 | **0.637878** | 0.596948 |
| vascular hilar enlargement | 1428 | **0.625607** | 0.60417 | 0.623934 |
| nodule | 736 | 0.446317 | 0.507929 | **0.546737** |
| air trapping | 1952 | 0.580408 | **0.631534** | 0.580882 |
| bullas | 192 | **0.744606** | 0.584846 | 0.486494 |
| ground glass pattern | 123 | **0.671321** | 0.661248 | 0.602802 |
| calcified adenopathy | 124 | **0.673757** | 0.624151 | 0.583562 |
| minor fissure thickening | 127 | 0.600411 | 0.558331 | **0.77315** |
| unchanged | 4036 | 0.618171 | **0.633874** | 0.395541 |
| clavicle fracture | 74 | 0.596974 | 0.596031 | **0.607514** |
| pseudonodule | 795 | 0.476977 | 0.472281 | **0.557981** |
| end on vessel | 63 | 0.397635 | 0.485072 | **0.560626** |

Table 2: **On average over 1000 bootstraps, the regularized model outperforms the unregularized and CheXzero models in fine-grained finding prediction.** Zero-shot AUCs for 57 padchest findings. The best AUC for each finding across the three tested models is shown in **bold**. As shown in Tab. 9 and Tab. 10, all winners are statistically significant except for CheXzero for the rib fracture finding.

## 5 Discussion & Limitations

In this work, we introduce a regularization method for contrastive language-image pre-trained models which encourages shrinkage of the image-patch and text-token similarities. We demonstrate how our regularization method can benefit zero-shot performance of these models by training a model that achieves SOTA zero-shot classification performance on a broad set of CXR findings. The improvements were robust across a wide range of tasks relative to many strong benchmarks, though in some instances the improvements were modest. Though our work was confined to a medical context, we believe it should be broadly applicable to many other areas where CLIP-style models are used, though these applications were beyond the scope of the present work. We believe our work contributes to a growing literature (Kumar et al., 2022; Mu et al., 2022; Meier et al., 2021) seeking to augment and improve CLIP-style models with inductive biases and domain-specific observations.

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

# A Appendix

## A.1 Regularization pseudocode

```python
def regularized_loss(Ims, Txts, lambda_patch, lambda_token):
    # image_encoder - ResNet-50, text_encoder - CXR-BERT-specialized
    # Ims[n, h, w, c], Txts[n, l] - minibatch of aligned images & texts
    # W_i[d_i, d_e] - learned projections of image patches to embed
    # W_t[d_t, d_e] - learned projections of text tokens to embed
    # P - number of image patches, T - number of text tokens

    # Setup; compute patch and token representations
    patch_f = image_encoder(Ims) #[n, d_i, P]
    token_f = text_encoder(Txts)  #[n, d_t, T]
    # project to joint embedding space [d_e] and normalize
    patch_e = l2_normalize(dot(patch_f, W_i), axis=1) #[n, d_e, P]
    token_e = l2_normalize(dot(token_f, W_t), axis=1) #[n, d_e, T]

    # CLIP Loss; Compute global embeddings
    image_e = l2_normalize(mean(patch_e, dim=2), axis=1) #[n, d_e]
    text_e = token_e[:, :, 0] #[n, d_e]
    # Compute scaled pairwise cosine similarity matrix
    clip_logits = dot(image_e, text_e.T) * exp(t) #[n, n]
    # Evaluate symmetric CLIP loss function
    labels = np.arange(n)
    loss_i = cross_entropy_loss(clip_logits, labels, axis=0)
    loss_t = cross_entropy_loss(clip_logits, labels, axis=1)
    clip_loss = (loss_i + loss_t)/2

    # Regularization; Compute patch-token similarity matrix
    sim_matrix = batch_multiply(token_e, patch_e) #[n, T, P]
    # Compute patch and token penalties
    patch_entropies = entropy(softmax(sim_matrix, axis = 2)) #[n, T]
    patch_penalty = lambda_patch * mean(patch_entropies)
    token_entropies = entropy(softmax(sim_matrix, axis = 1)) #[n, P]
    token_penalty = lambda_token * mean(token_entropies)

    regularized_loss = clip_loss + patch_penalty + token_penalty
    return regularized_loss
```

Figure 6: Pseudocode for our TIER regularization method. lambda_patch and lambda_token are hyperparameters that can be tuned depending on the desired level of patch/text-token sparsity.

## A.2    Additional heatmap examples

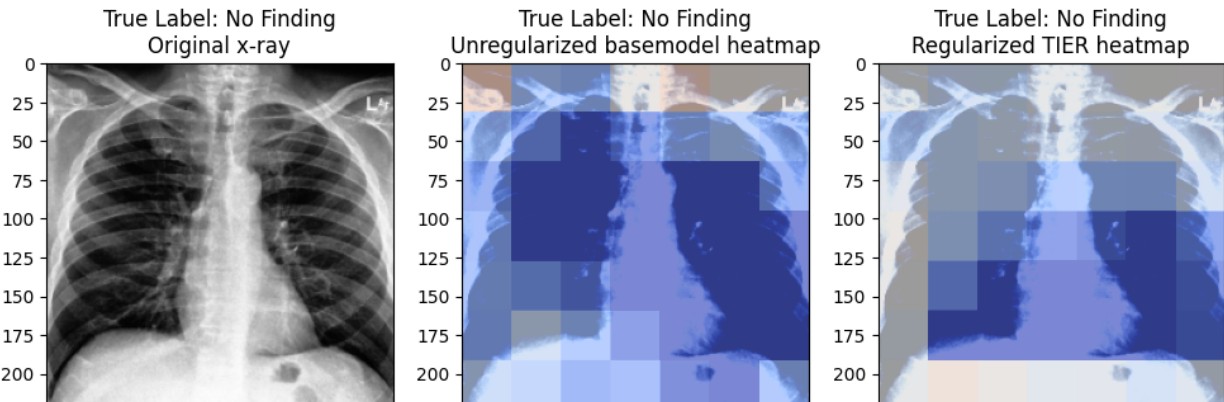

Figure 7: A CXR **negative** for cardiomegaly, overlaid with heatmaps displaying zero-shot cardiomegaly score for the unregularized (center) and regularized (right) models. Gray corresponds to a neutral (close to zero) zero-shot score, while red is a higher score and blue is a lower score.

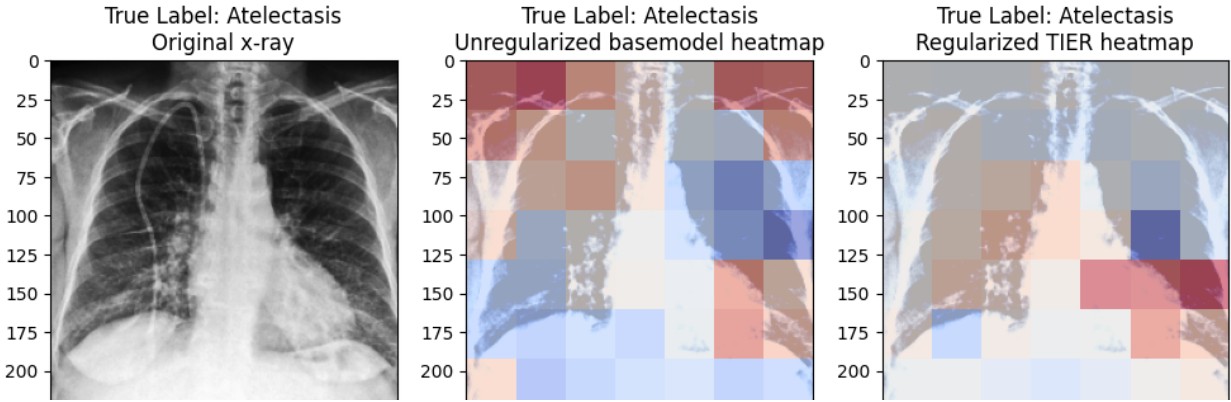

Figure 8: A CXR with atelectasis, overlayed with heatmaps displaying zero-shot atelectasis score for the unregularized (center) and regularized (right) models.

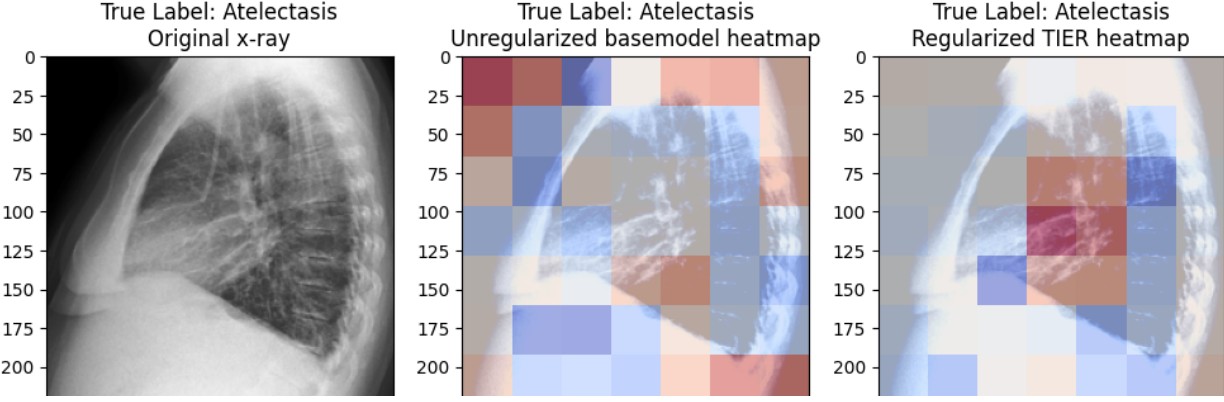

Figure 9: A lateral view of the previous atelectasis-positive CXR in Fig. 8, with zero-shot scores overlayed

## A.3 Hyperparameter Sweep

Here, we present the results of the hyperparameter sweep we used to select our lambda hyperparameters. Models were trained for a single epoch on MIMIC-CXR, and evaluated on the validation set of Chexpert. We use lambda values which maximized zero-shot AUC on the Chexpert validation set to train our regularized TIER model.

| $\lambda_p \mid \lambda_t$ | $\lambda_t = 0.00$ | $\lambda_t = 0.05$ | $\lambda_t = 0.10$ | $\lambda_t = 0.15$ | $\lambda_t = 0.20$ | $\lambda_t = 0.25$ |
|---|---|---|---|---|---|---|
| $\lambda_p = 0.00$ | 0.84708 | 0.84514 | 0.84624 | 0.84272 | 0.84870 | 0.83311 |
| $\lambda_p = 0.05$ | 0.85137 | 0.84661 | 0.84353 | 0.84542 | 0.84459 | 0.84712 |
| $\lambda_p = 0.10$ | 0.83971 | 0.85457 | 0.85059 | 0.84736 | 0.84464 | 0.83879 |
| $\lambda_p = 0.15$ | 0.83774 | 0.85367 | 0.85107 | 0.84174 | 0.83895 | 0.84242 |
| $\lambda_p = 0.20$ | 0.84399 | 0.85387 | **0.85469** | 0.84488 | 0.84747 | 0.83234 |
| $\lambda_p = 0.25$ | 0.84901 | 0.83306 | 0.84802 | 0.83939 | 0.83307 | 0.85160 |

Table 3: Zero-shot AUCs on the validation set after training a model with the given hyperparameters for 1 epoch on MIMIC-CXR. ($\lambda_p = 0.20$, $\lambda_t = 0.10$) were chosen to train our regularized TIER model.

## A.4 P-values for Chexpert evaluation

Here we present the p-values obtained from two-sample t-tests for differences in mean AUCs between TIER (Ours), CheXzero, and the unregularized baseline (Ours) for the chexpert evaluations.

| Label | CheXzero Mean | CheXzero Std | TIER (Ours) Mean | TIER Std | T statistic | P value |
|---|---|---|---|---|---|---|
| Average AUC | 0.893494 | 0.00696 | **0.903336** | 0.007213 | 31.05041349 | **4.5448E-173** |
| Cardiomegaly | 0.883397 | 0.013806 | **0.917135** | 0.010944 | 60.5584468 | **0** |
| Edema | 0.894235 | 0.015879 | **0.924225** | 0.01217 | 47.40345097 | **0** |
| Consolidation | **0.913176** | 0.014555 | 0.897116 | 0.023315 | -18.47763326 | **1.67297E-70** |
| Atelectasis | 0.842985 | 0.015325 | **0.865306** | 0.014135 | 33.85648614 | **5.547E-199** |
| Pleural Effusion | **0.933676** | 0.010558 | 0.912898 | 0.012155 | -40.81063492 | **2.4219E-265** |

Table 4: Two sample t-test for the difference of means between TIER (Ours) and CheXzero for n = 1000 bootstraps of the chexpert evaluation. All results are significant at the p=0.0001 level.

| Label | Unreg. (Ours) Mean | Unreg. Std | TIER (Ours) Mean | TIER Std | T statistic | P value |
|---|---|---|---|---|---|---|
| Average AUC | 0.897206 | 0.007486 | **0.903336** | 0.007213 | 18.64716405 | **1.13364E-71** |
| Cardiomegaly | 0.892394 | 0.012546 | **0.917135** | 0.010944 | 46.99391106 | **0** |
| Edema | 0.907286 | 0.013561 | **0.924225** | 0.01217 | 29.39763772 | **3.6269E-158** |
| Consolidation | **0.912127** | 0.024044 | 0.897116 | 0.023315 | -14.17328934 | **1.60039E-43** |
| Atelectasis | 0.857408 | 0.014522 | **0.865306** | 0.014135 | 12.32428706 | **1.08563E-33** |
| Pleural Effusion | **0.916813** | 0.011588 | 0.912898 | 0.012155 | -7.372034693 | **2.44822E-13** |

Table 5: Two sample t-test for the difference of means between TIER (ours) and the unregularized baseline (ours) for n = 1000 bootstraps of chexpert evaluation. All results are significant at the p=0.0001 level.

| Label | CheXzero Mean | CheXzero Std | Unreg. (Ours) Mean | Unreg. Std | T statistic | P value |
|---|---|---|---|---|---|---|
| Average AUC | 0.893494 | 0.00696 | **0.897206** | 0.007486 | 11.48385375 | **1.32063E-29** |
| Cardio | 0.883397 | 0.013806 | **0.892394** | 0.012546 | 15.25117349 | **9.17475E-50** |
| Edema | 0.894235 | 0.015879 | **0.907286** | 0.013561 | 19.7641868 | **1.47602E-79** |
| Consolidation | 0.913176 | 0.014555 | 0.912127 | 0.024044 | -1.180245623 | 0.238043032 |
| Atelectasis | 0.842985 | 0.015325 | **0.857408** | 0.014522 | 21.60293691 | **3.44293E-93** |
| Pleural Effusion | **0.933676** | 0.010558 | 0.916813 | 0.011588 | -34.01616352 | **1.7778E-200** |

Table 6: Two sample t-test for the difference of means between unregularized (ours) and CheXzero for n = 1000 bootstraps of chexpert evaluation. All results but consolidation are significant at the p=0.0001 level.

### A.5 Additional chexpert baselines

Here we display some additional baselines compared to our regularized TIER and unregularized baseline. In particular, we present the pretrained model we are using (Boecking et al., 2022) and CLIP (Radford et al., 2021).

| Label | TIER (Ours) | Unregularized (Ours) | BioViL | CLIP |
|---|---|---|---|---|
| Average | **0.903336** | 0.89721 | 0.63631 | 0.54056 |
| Cardiomegaly | **0.91714** | 0.89239 | 0.63300 | 0.56232 |
| Edema | **0.92423** | 0.90729 | 0.58706 | 0.5028 |
| Consolidation | 0.89712 | **0.91213** | 0.705898 | 0.6541 |
| Atelectasis | **0.86531** | 0.85741 | 0.58698 | 0.51719 |
| Pleural Effusion | 0.91290 | **0.91681** | 0.66864 | 0.46638 |

Table 7: The highest zero-shot AUC of the listed models is bolded. All models are performing zero-shot classification, meaning they have never explicitly been trained with any labels.

### A.6 Chexpert Queries

| Class Label | Caption |
|---|---|
| Cardiomegaly | Cardiomegaly is present. 
 The heart shadow is enlarged. 
 The cardiac silhouette is enlarged. |
| Pleural Effusion | Pleural Effusion is present. 
 Blunting of the costophrenic angles represents pleural effusions. 
 The pleural space is filled with fluid. 
 Layering pleural effusions are present. |
| Edema | Edema is present. 
 Increased fluid in the alveolar wall indicates pulmonary edema. |
| Consolidation | Consolidation is present. 
 Dense white area of right lung indicative of consolidation. |
| Atelectasis | Atelectasis is present. 
 Basilar opacity and volume loss is likely due to atelectasis. |
| No Finding | The lungs are clear. 
 No abnormalities are present. 
 The chest is normal. 
 No clinically significant radiographic abnormalities. 
 No radiographically visible abnormalities in the chest. |

Table 8: Query captions used for zero-shot classification. No Finding captions are used as the negative queries for classification on CheXpert, while the rest are used as positive queries for their respective labels.

## A.7 Chexpert Radiologist benchmarking

Here, we compare binary predictions from these zero-shot models against the predictions of 3 radiologists using MCC and F1 metrics. We selected optimal thresholds for MCC and F1 using their performance on the CheXpert validation set.

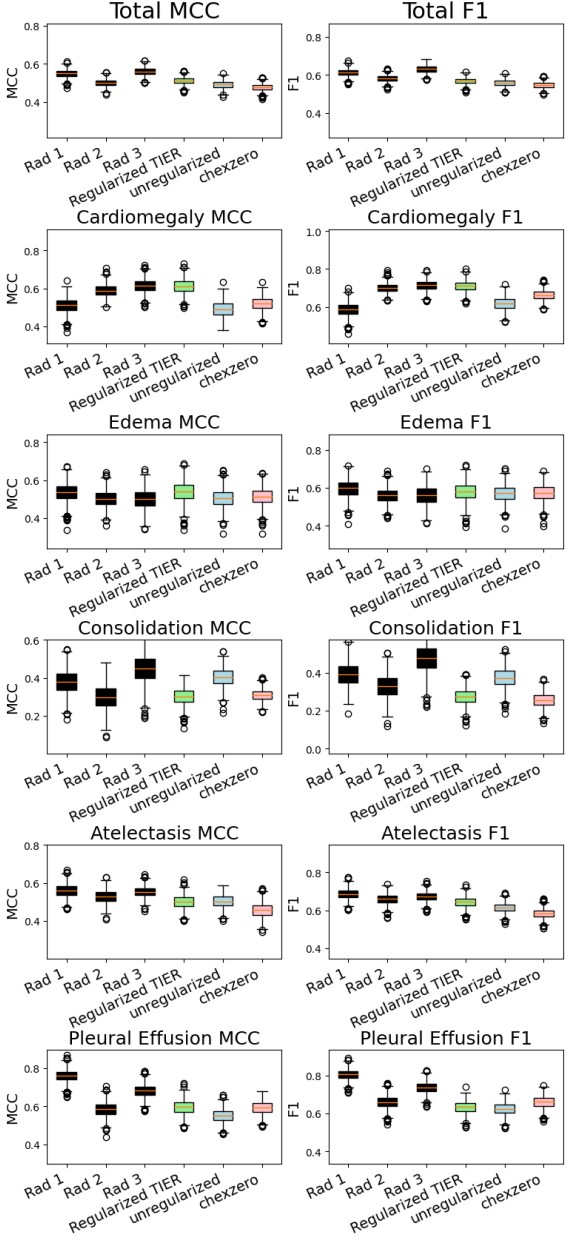

Figure 10: Benchmarking thresholded scores against radiologists. MCC (Matthews correlation coefficient) scores and F1 scores for the three zero-shot models and three radiologists are shown. These zero-shot models are competitive with radiologists for most labels. While TIER often outperform both the unregularized baseline at CheXzero on both metrics, it is equivalent or worse in several cases.

## A.8 P-values for padchest evaluation

Here we present the p-values obtained from two-sample t-tests for differences in mean AUCs for n = 1000 bootstraps between the various models for the Padchest evaluations.
Comparing CheXzero to TIER (Ours) on Padchest head-to-head:

| Label | CheXzero Mean | CheXzero Std | TIER (Ours) Mean | TIER Std | T statistic | P value |
|---|---|---|---|---|---|---|
| Average AUC | 0.726306 | 0.002558 | **0.75542** | 0.002347 | 265.2017626 | **0** |
| Number won | 21 | | **33** | | | |
| endotracheal tube | **0.98295** | 0.002315 | 0.979606 | 0.005919 | -16.63830145 | **2.56865E-58** |
| pleural effusion | **0.950519** | 0.002734 | 0.942045 | 0.002976 | -66.3097905 | **0** |
| pulmonary edema | **0.95646** | 0.008905 | 0.941289 | 0.008719 | -38.49466433 | **5.1755E-243** |
| heart insufficiency | 0.917819 | 0.004761 | **0.926097** | 0.005453 | 36.16180694 | **1.015E-220** |
| pulmonary fibrosis | 0.921793 | 0.008352 | **0.951654** | 0.0054 | 94.94482215 | **0** |
| cardiomegaly | **0.890467** | 0.002511 | 0.883692 | 0.002718 | -57.89827507 | **0** |
| vascular redistribution | 0.750592 | 0.018104 | **0.877236** | 0.013662 | 176.5761336 | **0** |
| consolidation | 0.865175 | 0.009475 | **0.878342** | 0.009453 | 31.10977089 | **1.3088E-173** |
| hilar congestion | 0.825707 | 0.007219 | **0.855435** | 0.008915 | 81.95062909 | **0** |
| pulmonary mass | 0.842056 | 0.01203 | 0.844107 | 0.012986 | 3.663919249 | 0.000254856 |
| cavitation | 0.853367 | 0.01575 | **0.857639** | 0.015601 | 6.093824451 | **1.31926E-09** |
| alveolar pattern | 0.763811 | 0.006956 | **0.87631** | 0.005049 | 413.89493 | **0** |
| calcified pleural thickening | 0.850707 | 0.019816 | **0.859651** | 0.020362 | 9.95447536 | **8.11361E-23** |
| lung metastasis | 0.827675 | 0.024452 | **0.877375** | 0.015578 | 54.20861509 | **0** |
| emphysema | **0.830578** | 0.009475 | 0.717841 | 0.013142 | -220.0452111 | **0** |
| interstitial pattern | 0.816432 | 0.005126 | **0.835144** | 0.005382 | 79.61342323 | **0** |
| costophrenic angle blunting | 0.69029 | 0.006777 | **0.769921** | 0.006022 | 277.7581639 | **0** |
| tuberculosis | 0.7978 | 0.027473 | **0.838961** | 0.020022 | 38.28895091 | **4.8878E-241** |
| atelectasis | **0.809232** | 0.009106 | 0.781707 | 0.008707 | -69.08700011 | **0** |
| reticular interstitial pattern | 0.822429 | 0.021108 | **0.844479** | 0.022699 | 22.49540647 | **4.4633E-100** |
| pneumonia | 0.773941 | 0.005566 | **0.813796** | 0.004699 | 173.0195883 | **0** |
| lobar atelectasis | 0.775147 | 0.017958 | **0.808411** | 0.014991 | 44.96696255 | **1.1767E-305** |
| normal | 0.753171 | 0.002546 | **0.776328** | 0.003632 | 165.0977062 | **0** |
| pleural thickening | 0.752537 | 0.016871 | **0.784428** | 0.014559 | 45.25503582 | **0** |
| reticulonodular interstitial pattern | 0.841414 | 0.027064 | **0.862346** | 0.024737 | 18.05301834 | **1.31327E-67** |
| infiltrates | **0.747836** | 0.006206 | 0.742854 | 0.006681 | -17.27714913 | **1.9175E-62** |
| hypoexpansion | 0.794564 | 0.014337 | **0.853423** | 0.011148 | 102.4871616 | **0** |
| hypoexpansion basal | 0.8018 | 0.015611 | **0.889652** | 0.013677 | 133.8542388 | **0** |
| humeral fracture | **0.749084** | 0.023222 | 0.742305 | 0.026582 | -6.07337803 | **1.49532E-09** |
| pneumothorax | **0.777442** | 0.018202 | 0.730643 | 0.026112 | -46.49431338 | **0** |
| multiple nodules | 0.716911 | 0.028257 | **0.790815** | 0.021245 | 66.10682237 | **0** |
| hyperinflated lung | **0.713202** | 0.018784 | 0.700879 | 0.018276 | -14.8691189 | **1.64612E-47** |
| bronchiectasis | 0.690117 | 0.01037 | **0.734643** | 0.009854 | 98.42837415 | **0** |
| adenopathy | **0.703924** | 0.02035 | 0.678726 | 0.016664 | -30.29508782 | **3.2002E-166** |
| mediastinal enlargement | **0.759299** | 0.022403 | 0.72538 | 0.026617 | -30.83087996 | **4.5077E-171** |
| laminar atelectasis | **0.679276** | 0.006312 | 0.67343 | 0.006914 | -19.74676158 | **1.97007E-79** |
| vertebral compression | 0.646448 | 0.025325 | **0.723955** | 0.018277 | 78.47811851 | **0** |
| rib fracture | 0.691037 | 0.020514 | 0.689835 | 0.022485 | -1.248835772 | 0.211871446 |
| tuberculosis sequelae | 0.584302 | 0.019415 | **0.796895** | 0.013529 | 284.0954171 | **0** |
| hilar enlargement | 0.678564 | 0.012333 | **0.721779** | 0.011469 | 81.14282586 | **0** |
| tracheal shift | **0.634359** | 0.019985 | 0.615827 | 0.019305 | -21.0906627 | **2.58516E-89** |
| mediastinal mass | 0.647695 | 0.034419 | **0.709825** | 0.031109 | 42.3483101 | **3.0928E-280** |
| central vascular redistribution | 0.354491 | 0.034431 | **0.728932** | 0.031302 | 254.4623065 | **0** |
| vertebral fracture | 0.499654 | 0.02921 | **0.791375** | 0.015662 | 278.3320678 | **0** |
| superior mediastinal enlargement | **0.596948** | 0.024252 | 0.551017 | 0.025206 | -41.52441949 | **2.9881E-272** |
| vascular hilar enlargement | 0.623934 | 0.007406 | **0.625607** | 0.007007 | 5.189078029 | **2.32751E-07** |
| nodule | **0.546737** | 0.010556 | 0.446317 | 0.010124 | -217.1150565 | **0** |
| air trapping | 0.580882 | 0.006245 | 0.580408 | 0.005897 | -1.745118047 | 0.081118057 |
| bullas | 0.486494 | 0.020169 | **0.744606** | 0.018356 | 299.2954863 | **0** |
| ground glass pattern | 0.602802 | 0.027608 | **0.671321** | 0.020656 | 62.84105487 | **0** |
| calcified adenopathy | 0.583562 | 0.02315 | **0.673757** | 0.019228 | 94.77745188 | **0** |
| minor fissure thickening | **0.77315** | 0.018481 | 0.600411 | 0.025956 | -171.4357871 | **0** |
| unchanged | 0.395541 | 0.004386 | **0.618171** | 0.004502 | 1120.102208 | **0** |
| clavicle fracture | **0.607514** | 0.036746 | 0.596974 | 0.037946 | -6.309943613 | **3.42827E-10** |
| pseudonodule | **0.557981** | 0.009991 | 0.476977 | 0.011371 | -169.2291715 | **0** |
| end on vessel | **0.560626** | 0.035794 | 0.397635 | 0.041243 | -94.38338959 | **0** |
| COPD signs | **0.751217** | 0.003745 | 0.650859 | 0.004075 | -573.421393 | **0** |

Table 9: Two sample t-test for the difference of means for our padchest evaluation. In this table, we compare the previous SOTA, CheXzero, with our regularized model, TIER. Each model had AUC evaluated on n = 1000 bootstraps of the radiologists-labeled subset of padchest. All but the air trapping, rib fracture, and pulmonary mass results are significant at the p=0.0001 level.

Comparing Unregularized baseline (Ours) to TIER (Ours) on Padchest head-to-head:

| Label | Unreg. (Ours) Mean | Unreg. Std | TIER (Ours) Mean | TIER Std | T statistic | P value |
|---|---|---|---|---|---|---|
| Average AUC | 0.742534 | 0.002567 | **0.75542** | 0.002347 | 117.1556311 | **0** |
| Number won | 23 | | **30** | | | |
| endotracheal tube | 0.956634 | 0.008621 | **0.979606** | 0.005919 | 69.46676879 | **0** |
| pleural effusion | 0.930292 | 0.003377 | **0.942045** | 0.002976 | 82.56984314 | **0** |
| pulmonary edema | **0.945415** | 0.008526 | 0.941289 | 0.008719 | -10.69926209 | **5.11898E-26** |
| heart insufficiency | **0.927177** | 0.005282 | 0.926097 | 0.005453 | -4.498643855 | **7.23167E-06** |
| pulmonary fibrosis | 0.944147 | 0.006584 | **0.951654** | 0.0054 | 27.87855697 | **9.3138E-145** |
| cardiomegaly | 0.883339 | 0.002701 | 0.883692 | 0.002718 | 2.913187348 | 0.00361733 |
| vascular redistribution | 0.872019 | 0.013502 | **0.877236** | 0.013662 | 8.588841321 | **1.73859E-17** |
| consolidation | 0.849872 | 0.011331 | **0.878342** | 0.009453 | 61.01092612 | **0** |
| hilar congestion | 0.850243 | 0.008691 | **0.855435** | 0.008915 | 13.18723777 | **3.87838E-38** |
| pulmonary mass | **0.872299** | 0.012346 | 0.844107 | 0.012986 | -49.75455492 | **0** |
| cavitation | 0.794295 | 0.017229 | **0.857639** | 0.015601 | 86.18194266 | **0** |
| alveolar pattern | 0.816974 | 0.006339 | **0.87631** | 0.005049 | 231.535272 | **0** |
| calcified pleural thickening | 0.84287 | 0.019733 | **0.859651** | 0.020362 | 18.71497205 | **3.84423E-72** |
| lung metastasis | 0.860837 | 0.017219 | **0.877375** | 0.015578 | 22.52272401 | **2.7303E-100** |
| emphysema | 0.718377 | 0.013149 | 0.717841 | 0.013142 | -0.911743481 | 0.362013758 |
| interstitial pattern | **0.840368** | 0.005014 | 0.835144 | 0.005382 | -22.45845998 | **8.6714E-100** |
| costophrenic angle blunting | **0.808131** | 0.00461 | 0.769921 | 0.006022 | -159.323741 | **0** |
| tuberculosis | **0.843741** | 0.024721 | 0.838961 | 0.020022 | -4.751556022 | **2.1624E-06** |
| atelectasis | **0.791507** | 0.0086 | 0.781707 | 0.008707 | -25.32275656 | **6.2416E-123** |
| reticular interstitial pattern | **0.867637** | 0.019332 | 0.844479 | 0.022699 | -24.56163626 | **1.2365E-116** |
| pneumonia | 0.796614 | 0.005068 | **0.813796** | 0.004699 | 78.6172414 | **0** |
| lobar atelectasis | **0.815725** | 0.013775 | 0.808411 | 0.014991 | -11.36063996 | **4.99873E-29** |
| normal | **0.790588** | 0.003023 | 0.776328 | 0.003632 | -95.42795587 | **0** |
| pleural thickening | 0.754608 | 0.016022 | **0.784428** | 0.014559 | 43.55866303 | **5.5682E-292** |
| reticulonodular interstitial pattern | 0.838374 | 0.026992 | **0.862346** | 0.024737 | 20.70489002 | **1.95964E-86** |
| infiltrates | 0.735399 | 0.006646 | **0.742854** | 0.006681 | 25.01662683 | **2.1878E-120** |
| hypoexpansion | **0.871452** | 0.010845 | 0.853423 | 0.011148 | -36.65734018 | **1.9617E-225** |
| hypoexpansion basal | 0.874477 | 0.014044 | **0.889652** | 0.013677 | 24.47917337 | **5.8658E-116** |
| humeral fracture | 0.672935 | 0.028067 | **0.742305** | 0.026582 | 56.74716788 | **0** |
| pneumothorax | 0.728547 | 0.021418 | 0.730643 | 0.026112 | 1.962595625 | 0.049831759 |
| multiple nodules | **0.852951** | 0.020427 | 0.790815 | 0.021245 | -66.66997483 | **0** |
| hyperinflated lung | 0.667704 | 0.017846 | **0.700879** | 0.018276 | 41.06987413 | **7.5179E-268** |
| bronchiectasis | **0.743998** | 0.009203 | 0.734643 | 0.009854 | -21.94072646 | **8.91339E-96** |
| adenopathy | **0.73105** | 0.017982 | 0.678726 | 0.016664 | -67.49145771 | **0** |
| mediastinal enlargement | 0.666796 | 0.028092 | **0.72538** | 0.026617 | 47.87154657 | **0** |
| laminar atelectasis | **0.687839** | 0.006426 | 0.67343 | 0.006914 | -48.27281362 | **0** |
| vertebral compression | **0.734413** | 0.019904 | 0.723955 | 0.018277 | -12.2383364 | **2.91558E-33** |
| rib fracture | 0.668069 | 0.023081 | **0.689835** | 0.022485 | 21.36070437 | **2.38093E-91** |
| tuberculosis sequelae | 0.773832 | 0.014015 | **0.796895** | 0.013529 | 37.44003048 | **6.6415E-233** |
| hilar enlargement | 0.714687 | 0.011757 | **0.721779** | 0.011469 | 13.65450364 | **1.19318E-40** |
| tracheal shift | 0.500734 | 0.02327 | **0.615827** | 0.019305 | 120.3743682 | **0** |
| mediastinal mass | 0.409473 | 0.031929 | **0.709825** | 0.031109 | 213.0621769 | **0** |
| central vascular redistribution | 0.567387 | 0.046306 | **0.728932** | 0.031302 | 91.39738689 | **0** |
| vertebral fracture | **0.86009** | 0.013028 | 0.791375 | 0.015662 | -106.6628934 | **0** |
| superior mediastinal enlargement | **0.637878** | 0.021595 | 0.551017 | 0.025206 | -82.75530027 | **0** |
| vascular hilar enlargement | 0.60417 | 0.007237 | **0.625607** | 0.007007 | 67.29618289 | **0** |
| nodule | **0.507929** | 0.011538 | 0.446317 | 0.010124 | -126.9283034 | **0** |
| air trapping | **0.631534** | 0.005968 | 0.580408 | 0.005897 | -192.699814 | **0** |
| bullas | 0.584846 | 0.023316 | **0.744606** | 0.018356 | 170.2487729 | **0** |
| ground glass pattern | 0.661248 | 0.021925 | **0.671321** | 0.020656 | 10.57463045 | **1.81468E-25** |
| calcified adenopathy | 0.624151 | 0.023153 | **0.673757** | 0.019228 | 52.12228786 | **0** |
| minor fissure thickening | 0.558331 | 0.022571 | **0.600411** | 0.025956 | 38.68594962 | **7.5149E-245** |
| unchanged | **0.633874** | 0.004591 | 0.618171 | 0.004502 | -77.22708382 | **0** |
| clavicle fracture | 0.596031 | 0.041522 | 0.596974 | 0.037946 | 0.530145578 | 0.596069909 |
| pseudonodule | 0.472281 | 0.010954 | **0.476977** | 0.011371 | 9.405369698 | **1.37144E-20** |
| end on vessel | **0.485072** | 0.037602 | 0.397635 | 0.041243 | -49.54199726 | **0** |
| COPD signs | **0.652912** | 0.00394 | 0.650859 | 0.004075 | -11.45351594 | **1.83491E-29** |

Table 10: Two sample t-test for the difference of means for our padchest evaluation. In this table, we compare the previous SOTA, CheXzero, with our regularized model, TIER. Each model had AUC evaluated on n = 1000 bootstraps of the radiologists-labeled subset of padchest. All but the cardiomegaly, emphysema, pneumothorax, and clavicle fracture findings are significant at the p=0.0001 level.

### A.9 Code/model availability

Code is available at [https://github.com/apalepu13/TIER_Regularized_CLIP](https://github.com/apalepu13/TIER_Regularized_CLIP). For model checkpoints for the regularized/unregularized/fully supervised models, contact the authors. The BioViL model is available at [https://huggingface.co/microsoft/BiomedVLP-CXR-BERT-specialized](https://huggingface.co/microsoft/BiomedVLP-CXR-BERT-specialized).

