# OpenReview forum: "TIER: Text-Image Entropy Regularization for CLIP-style models"
_TMLR — Rejected by TMLR_

### Review · Reviewer_EZZr · 2022-12-23

**Summary Of Contributions:**

This work aims to regularize the CLIP-style model to be only sensitive to the paired image-text tokens, which has not been considered in the previous CLIP based pre-training. The image-text tokens are penalized via entropy minimization. The experimental results demonstrate the effectiveness of the proposed approach.

**Audience:**

Yes

**Claims And Evidence:**

No

**Requested Changes:**

1. The novelty of the proposed method should be further highlighted, with comparison to the previous works [1, 2]. These previous works both focus on promoting the alignment between the patch-like regions in an image and the text tokens. I can not learn the major contribution of the proposed method with these previous works.
2. The motivation of utilizing the entropy minimization for better patch-text alignment should be clarified more clearly. From my view, it is highly similar with the contrastive objective used in CLIP-like model training. So, if simple entropy-based minimization can only bring incremental improvement.
3. If it is reasonable as you take single-patch-based patch-text pairs for entropy minimization? It is obvious from the Chest X-Ray image that maybe some image patches should be aligned to the same text token. With every single patch aligned to different text tokens, there maybe disagreement between similar image patches.
4. The experimental results show very marginal improvements compared with previous SOTA. Moreover, these experimental results are obtained from a single run, which is not convincing. As the performance improvements are very incremental, multiple runs of the experiments are highly required here, so that mean and std can be reported.
5. The visualization results in Figure 3 look not good. With the penalty term, the learned model is still paying attention to the non-lung areas, which is opposite to this work’s aim (better aligning the truly paired patch-text pairs). Also, if you could show more totally failure samples, which can help us better understand how the framework works.
6. In Figure 7, for most cases, the F1 score of the proposed method is almost the same with the one of the baseline without regularization, and sometimes the proposed PITER underperforms the baseline method. Could you please give more convincing explanations on these?
7. Possible typos: 1) 2nd para. in intro., ‘vision transformer (?)’ missing the citation.

**Strengths And Weaknesses:**

Strengths:
1. The observation is good that the image-text pairs are aligned coarsely. Such too coarse alignment may not be effective enough facing various downstream tasks.
2. The proposed regularization term can be applied to any CLIP-like models without further modifications regarding the model architecture.

Weaknesses:
1. The novelty of the proposed method is very limited. As the patch/token-based alignment has been well studied before, such as GLIP [1] and RegionCLIP [2]. The proposed regularization term is more like a trick.
2. The improvements regarding the reported experimental results are very marginal, especially in Table 2 (only 0.007 average AUC improvement).

[1] Li LH, Zhang P, Zhang H, Yang J, Li C, Zhong Y, Wang L, Yuan L, Zhang L, Hwang JN, Chang KW. Grounded language-image pre-training. In Proceedings of the IEEE/CVF Conference on Computer Vision and Pattern Recognition 2022 (pp. 10965-10975).
[2] Zhong Y, Yang J, Zhang P, Li C, Codella N, Li LH, Zhou L, Dai X, Yuan L, Li Y, Gao J. Regionclip: Region-based language-image pretraining. In Proceedings of the IEEE/CVF Conference on Computer Vision and Pattern Recognition 2022 (pp. 16793-16803).

---

> ### Author Response · Authors · 2023-02-07
> **Response to Reviewer EZZr (part 1)**
>
> Please see the high level summary of response to reviewers for our general response. Here, we will try to address the specific points raised in your review.
>
> **Weaknesses:**
> > The novelty of the proposed method is very limited. As the patch/token-based alignment has been well studied before, such as GLIP [1] and RegionCLIP [2]. The proposed regularization term is more like a trick.
>
> We have added a section discussing these approaches, as well as several others, on page 2 of our manuscript.
>
> GLIP and RegionCLIP both rely on separate object detection models. These object detection methods, while fairly effective for natural images, have not been equivalently utilized in medical domains, and indeed these references were implemented for natural images. Our method is a simple and lightweight addition to any CLIP-style architecture. Our method may also be desirable in cases where the labels of interest aren’t located in contiguous image regions; for example, if a certain finding is spread across in distinct locations in the lung. This would require “sparsity” but would not be easily captured by localized object detection.
>
> > The improvements regarding the reported experimental results are very marginal, especially in Table 2 (only 0.007 average AUC improvement).
>
> As we describe in our high-level summary, we re-ran our experiments with 1000 bootstraps using the single best checkpoint to ensure that our AUC results were consistent on both the CheXpert and Padchest evaluations.
>
> We indeed find that the average AUC improvement we see with our regularized model is consistent and statistically significant at a p=0.0001 level for both evaluations. Additionally, our model has superior performance on a majority of findings when compared head-to-head with the CheXzero and unregularized models, as seen by Tables 4,5,6, and Tables 9,10 in the appendix (pages 15, 18, and 19).
> We do concede that many of these improvements are modest, and that we do not have superior performance for every finding. That being said, even modest gains may be extremely important for high-stakes scenarios such as medicine, and our models are overall an advance on zero-shot interpretation of chest X-rays.
>
> **Requested Changes**:
> > The novelty of the proposed method should be further highlighted, with comparison to the previous works [1, 2]. These previous works both focus on promoting the alignment between the patch-like regions in an image and the text tokens. I can not learn the major contribution of the proposed method with these previous works.
>
> We have added a section discussing these prior methods, as well as several others, on page 2 of our manuscript.
> GLIP and RegionCLIP both rely on separate object detection models. These object detection methods, while fairly effective for natural images, have not been equivalently utilized in medical domains, and indeed these approaches were implemented for natural images.
> Additionally, our method is a simple and lightweight addition to any CLIP-style architecture that has some notion of “local embeddings”. Our method may also be desirable in cases where the labels of interest aren’t located in contiguous image regions; for example, if a certain finding is spread across in distinct locations in the lung. This would still require “sparsity” but would not be easily captured by an individual bounding boxes.
>
> > The motivation of utilizing the entropy minimization for better patch-text alignment should be clarified more clearly. From my view, it is highly similar with the contrastive objective used in CLIP-like model training. So, if simple entropy-based minimization can only bring incremental improvement.
>
> The contrastive objective with CLIP aims to align our global image representation with our global text representation. The entropy minimization portion serves to encourage the model, while learning these global representations, to also try to sparsify the patch similarities to each text token and the text token similarities to each patch. This additional regularization should ideally encourage the model to consolidate its activation to the most relevant image patches during zero-shot classification.
>
> **References**
> - [1] Li LH, Zhang P, Zhang H, Yang J, Li C, Zhong Y, Wang L, Yuan L, Zhang L, Hwang JN, Chang KW. Grounded language-image pre-training. In Proceedings of the IEEE/CVF Conference on Computer Vision and Pattern Recognition 2022 (pp. 10965-10975).
> - [2] Zhong Y, Yang J, Zhang P, Li C, Codella N, Li LH, Zhou L, Dai X, Yuan L, Li Y, Gao J. Regionclip: Region-based language-image pretraining. In Proceedings of the IEEE/CVF Conference on Computer Vision and Pattern Recognition 2022 (pp. 16793-16803).

---

> ### Author Response · Authors · 2023-02-07
> **Response to Reviewer EZZr (part 2)**
>
> **Requested Changes (continued)**
> > If it is reasonable as you take single-patch-based patch-text pairs for entropy minimization? It is obvious from the Chest X-Ray image that maybe some image patches should be aligned to the same text token. With every single patch aligned to different text tokens, there maybe disagreement between similar image patches.
>
> Tuning the lambda hyperparameters allows us to avoid this type of situation. In particular, we place a stronger emphasis on each text token being represented by a few image patches (lambda_patch), and relatively less weight on each patch being represented by few text tokens (lambda_text), meaning we are less likely to encounter the issue you describe.
>
> Additionally, the model must still achieve  a low contrastive loss in addition to entropy minimization. Thus, if aligning every patch to different text tokens would significantly undermine the model’s ability to align global embeddings, then the model will take on additional entropy in order to lower its contrastive loss.
>
>
> > The experimental results show very marginal improvements compared with previous SOTA. Moreover, these experimental results are obtained from a single run, which is not convincing. As the performance improvements are very incremental, multiple runs of the experiments are highly required here, so that mean and std can be reported.
>
> As we describe in our high-level summary, we re-ran our experiments with 1000 bootstraps using the single best checkpoint to ensure that our AUC results were consistent on both the CheXpert and Padchest evaluations.
>
> We indeed find that the average AUC improvement we see with our regularized model is consistent and statistically significant at a p=0.0001 level for both evaluations. Additionally, our model has superior performance on a majority of findings when compared head-to-head with the CheXzero and unregularized models, as seen by Tables 4,5,6 and Tables 9,10 in the appendix (pages 15, 18, and 19).
> We do concede that many of these improvements are modest, and that we do not have superior performance for every finding. That being said, even modest gains may be extremely important for high-stakes scenarios such as medicine, and our models are overall an advance on zero-shot interpretation of chest X-rays.
>
> > The visualization results in Figure 3 look not good. With the penalty term, the learned model is still paying attention to the non-lung areas, which is opposite to this work’s aim (better aligning the truly paired patch-text pairs). Also, if you could show more totally failure samples, which can help us better understand how the framework works.
>
> I think the visualization in what was formerly figure 3 (and now is figure 7) do show a significant improvement between the unregularized and regularized models. The model is being queried for Cardiomegaly, which is not present in the image. Thus, it is desirable for the model to have negative (blue) zero-shot scores in regions near  the heart. The regularized model does still pay some attention to non-heart regions, but the regions it focuses on are far more localized to the heart when compared to the unregularized model.
> We added additional examples of heatmaps (fig 2,3,7,8,9) which should help support our claim that the regularization improves the focus of the model.
>
> > In Figure 7, for most cases, the F1 score of the proposed method is almost the same with the one of the baseline without regularization, and sometimes the proposed PITER underperforms the baseline method. Could you please give more convincing explanations on these?
>
> After repeating this experimentation with 1000 bootstraps, we can see in figure 10 (page 17 of the appendix) that the TIER model consistently outperforms the unregularized baseline and CheXzero on average and on a majority of findings. Like with the AUC evaluation, we see exceptions to this with the consolidation and pleural effusion findings.
>
> > Possible typos: 1) 2nd para. in intro., ‘vision transformer (?)’ missing the citation.
>
> We have fixed these errors.

---

### Review · Reviewer_ZJ8u · 2023-01-05

**Summary Of Contributions:**

In this paper, the authors proposed a regularization scheme on contrastive language-image pre-trained (CLIP) models. In particular, the proposed regularization penalizes the entropy of text token-image patch similarities for each text token, and the entropy of image patch-text token similarities for each image patch. The authors trained CLIP models with the proposed regularization on chest X-ray images, and found that the CLIP models trained with the proposed entropy regularization achieved competitive zero-shot classification results on test datasets compared to the CLIP models trained without such regularization and the state-of-the-art Chexzero / MedCLIP model.

**Audience:**

Yes

**Broader Impact Concerns:**

There are no concerns on the ethical implications of the work that would require adding a Broader Impact Statement.

**Claims And Evidence:**

Yes

**Requested Changes:**

- Include more examples (and possibly a human study) to show that the regularized model is indeed better at focusing on clinically justifiable regions than the unregularized model.
- Add the missing references.
- It is not necessary to include Python code for the proposed regularization (Figure 2).
- Work on some additional techniques that can further improve the zero-shot classification performance.

**Strengths And Weaknesses:**

Strengths:
- The paper is clearly written and easy to follow.
- The proposed regularization is easy to implement, and in some cases, does seem to improve the zero-shot classification performance.

Weaknesses:
- The experimental evaluations of the proposed regularization are weak. For example, the authors only presented two pairs of heatmaps, produced by CLIP models trained with and without the proposed regularization, to show that the model trained with regularization does seem to focus on "clinically justifiable" regions. It would be nice to have more examples (even a human study) to show that this is generally the case.
- The improvement in zero-shot classification performance from the proposed regularization is not obvious -- for a lot of labels, the unregularized model performed similarly as the regularized model.
- Some references are missing, e.g., vision transformer (page 1) and MedCLIP (page 8).

---

> ### Author Response · Authors · 2023-02-07
> **Response to Reviewer ZJ8u**
>
> Please see the high level summary of response to reviewers for our general response. Here, we will try to address the specific points raised in your review.
>
> **Weaknesses**
> > The experimental evaluations of the proposed regularization are weak. For example, the authors only presented two pairs of heatmaps, produced by CLIP models trained with and without the proposed regularization, to show that the model trained with regularization does seem to focus on "clinically justifiable" regions. It would be nice to have more examples (even a human study) to show that this is generally the case.
>
> Fig 5 and 6 demonstrate that regularization works as intended across a larger set of images. These figures demonstrate that across a large set of images, the regularized model has fewer patches with high similarities to the text embedding, implying that regularization induces sparsity of the image patches.
>
> While this does demonstrate the similarities are sparser, it does not inherently prove that the patches being utilized are more “clinically justifiable”. A human study would be most beneficial to support this claim, but we are unable to provide such results at this time. That being said, we do provide additional examples of zero-shot heatmaps in the appendix (fig 7,8,9 on page 14) to better support our claim that the activated regions are clinically justifiable. In these heatmaps, we see that the regularized model tends to be more centrally focused on patches near the lungs and heart, whereas the unregularized model often has extreme zero-shot scores on patches on the borders of the image that are clearly clinically irrelevant.
>
> > The improvement in zero-shot classification performance from the proposed regularization is not obvious -- for a lot of labels, the unregularized model performed similarly as the regularized model.
>
> As we describe in our high-level summary, we re-ran our experiments with 1000 bootstraps using the single best checkpoint to ensure that our AUC results were consistent on both the CheXpert and Padchest evaluations.
>
> We indeed find that the average AUC improvement we see with our regularized model is consistent and statistically significant at a p=0.0001 level for both evaluations. Additionally, our model has superior performance on a majority of findings when compared head-to-head with the CheXzero and unregularized models, as seen by Tables 4,5,6, and Tables 9,10 in the appendix (pages 15, 18, and 19).
> We do concede that many of these improvements are modest, and that we do not have superior performance for every finding. That being said, even modest gains may be extremely important for high-stakes scenarios such as medicine, and our models are overall an advance on zero-shot interpretation of chest X-rays.
>
> > Some references are missing, e.g., vision transformer (page 1) and MedCLIP (page 8).
>
> We have added these missing references to the paper.
>
> **Requested Changes**
> > Include more examples (and possibly a human study) to show that the regularized model is indeed better at focusing on clinically justifiable regions than the unregularized model.
>
> We have included several additional examples of these heatmaps in the appendix.
>
> > Add the missing references.
>
> We have fixed these missing references.
>
> > It is not necessary to include Python code for the proposed regularization (Figure 2).
>
> We moved this pseudocode to the appendix.
>
> > Work on some additional techniques that can further improve the zero-shot classification performance.
>
> While we aim to eventually pursue this, we believe these changes are outside the scope of our current project focusing on our regularization technique.

---

### Review · Reviewer_bicQ · 2023-01-07

**Summary Of Contributions:**

This paper presents a new learning strategy to further improve the popular vision-language pre-trained model CLIP. In particular, authors propose to include two regularization terms on the text-patch embeddings, motivated by the fact that, for a given text prompt, only a few patches should be highly correlated (and vice versa for patch to text). To enforce this scenario, authors integrate a term that minimizes the Shannon entropy of the softmax probabilities of each row (or column) for the different correlations (i.e., patch-to-text and text-to-patch). To evaluate the proposed approach, authors resort to two classification benchmarks (CheXpert and PadChest) and compare to several existing works.

**Audience:**

Yes

**Broader Impact Concerns:**

No concerns.

**Claims And Evidence:**

No

**Requested Changes:**

Please see my previous comments.

**Strengths And Weaknesses:**

### Strengths

- The paper is easy to read and the methodology is sound.

- Vision Language pre-training models are an emerging topic, which is gaining popularity, thus, the addressed problem is interesting and relevant.

### Weaknesses

- Contribution is somehow limited, and its motivation is questionable.

- The empirical validation is unconvincing.

- Many related works in computer vision are missing in the literature review and validation.

Please find my detailed comments below.

### 1. Contribution.

- 1.1. The proposed methodology basically extends the work in Boecking et al. by adding two regularization terms, whose motivation/effect is questionable (see more details about this in the next point).

- 1.2. These two terms minimize the Shannon entropy of the softmax predictions for each row (column). The motivation behind this is that for a given text, only a few patches should be highly correlated, and vice-versa. While this might be somehow true for positive instances, in the case of negative cases (e.g., no disease), the activation of all the patches should be homogeneous. If we take the case of the correlation between patches and a given text prompt, the optimal solution that minimizes the entropy is similar to one-hot encoding, i.e., all the mass goes to one single patch, which will give incorrect solutions/distributions for images without diseases. In addition, even when a disease or pathological region is present in an image, it might cover a significant portion of the image. Thus, I find the motivation behind these two terms quite weak.

- 1.3. Another claimed contribution is novel state-of-the-art results on the zero-shot task (Chezxero in Table 2). Nevertheless, the differences wrt the unregularized model are very marginal, which makes me wonder about the real impact of the proposed methodology.

### 2. Methodology.

- 2.1. The methodology is mathematically correct, despite my concerns raised in point 1.2.

- 2.2. Authors state that the proposed terms in equation 1) and 2) are weighted by two hyper parameters ($\lambda_p$ and $\lambda_t$). Nevertheless, the code given is incorrect, as these hyperparameters are not present in the provided code. I wonder whether these hyperparameters are indeed used.

### 3. Results.

- 3.1. In terms of results, the presented empirical validation is not convincing. The average improvement compared to the unregularized model is marginal (specially in Table 2).

- 3.2. The list of included methods in the experiments is insufficient. There exist many recent approaches in the computer vision literature that must be included to assess the real value of the proposed approach. See for example [a-e]. These works should also be included in the literature review and properly discussed.

- 3.3. Ablation study on the impact of the different values of the $\lambda_p$ and $\lambda_t$ hyperparameters should be included too.

- 3.4. I wonder why the patch size used in this work (Figures 3 and 4) is different from the size used in Boecking et al. If the size is different, authors should assess the impact of the different patch sizes and motivate why they employed a different size.

### References

- [a] Huang SC, Shen L, Lungren MP, Yeung S. Gloria: A multimodal global-local representation learning framework for label-efficient medical image recognition. ICCV’21

- [b] Yao L, Huang R, Hou L, Lu G, Niu M, Xu H, Liang X, Li Z, Jiang X, Xu C. FILIP: Fine-grained Interactive Language-Image Pre-Training. ICLR’21

- [c] Li J, HE X, Wei L, Qian L, Zhu L, Xie L, Zhuang Y, Tian Q, Tang S. Fine-Grained Semantically Aligned Vision-Language Pre-Training. NeurIPS’22

- [d] Zhong Y, Yang J, Zhang P, Li C, Codella N, Li LH, Zhou L, Dai X, Yuan L, Li Y, Gao J. Regionclip: Region-based language-image pretraining. CVPR’22

- [e] Wang F, Zhou Y, Wang S, Vardhanabhuti V, Yu L. Multi-Granularity Cross-modal Alignment for Generalized Medical Visual Representation Learning. NeurIPS’22

---

> ### Author Response · Authors · 2023-02-07
> **Response to Reviewer bicQ (part 1)**
>
> Please see the high level summary of response to reviewers for our general response. Here, we will try to address the specific points raised in your review.
>
> > 1.1. The proposed methodology basically extends the work in Boecking et al. by adding two regularization terms
>
> Boecking et al. developed a CLIP-style architecture with local (patch) embeddings. We employ several changes to produce our unregularized model, as described in the high level summary, and find significant performance improvements as seen in table 7 of the appendix (page 16). Our regularized model differs from our unregularized model only with our addition of the two regularization terms described in the paper.
>
> > 1.2 “...in the case of negative cases (e.g., no disease), the activation of all the patches should be homogeneous.”
>
> We contend that even in negative cases, the activation of all patches should ideally not be homogenous.
> Consider an example image that has no finding present, and suppose we query the image for “No finding is present”. Rather than being homogenous, the areas where findings typically should present themselves ideally would have a higher similarity to our query than typically irrelevant patches. In particular, the high similarity regions being more concentrated in the lungs/lower chest is more desirable than high similarity being spread broadly over the entire image for CXRs being queried for “No finding”. This is because the lungs & lower chest are regions where clinical findings are more often identified.
>
> > 1.2 “...If we take the case of the correlation between patches and a given text prompt, the optimal solution that minimizes the entropy is similar to one-hot encoding, i.e., all the mass goes to one single patch, which will give incorrect solutions/distributions for images without diseases.”
>
> Our regularization terms are only a portion of the overall loss, which is still mostly dictated by global image-text alignment via the standard CLIP loss. Thus, while on net our terms do “sparsify” the mass to fewer patches as desired, the model will not focus on a single patch if doing so would compromise its ability to correctly align images and text. Tuning of our hyperparameters (“lambda_patch”, “lambda_text”), allows us to adjust the relative strength of our regularization; lowering these hyperparameters can avoid the type of errant solution you describe.
>
> > 1.2 “...In addition, even when a disease or pathological region is present in an image, it might cover a significant portion of the image.”
>
> Even in a setting where important regions present themselves across a significant portion of the image, the lambda hyperparameters can simply be lowered or set to 0 to diminish/eliminate the effect of regularization. That being said, CXR clinical findings tend to be localized. Consider the 5 CheXpert findings: Cardiomegaly, Pleural Effusion, Consolidation, Atelectasis, and Edema:
> Cardiomegaly tends to be localized around the heart. Pleural effusion tends to be localized at the base of a lung. Consolidation is typically identified in particular location(s) within the lung. Atelectasis can be identified by a localized portion of a lung lobe that is collapsed. Edema is perhaps the least localized of these findings, but is still seen throughout the lungs, and not in the various other regions of a CXR. Thus, all of these findings could arguably benefit by encouraging more sparse representations.
>
> > 1.3 "...the differences wrt the unregularized model are very marginal, which makes me wonder about the real impact of the proposed methodology."
>
> As we describe in our high-level summary, we re-ran our experiments with 1000 bootstraps using the single best checkpoint to ensure that our AUC results were consistent on both the CheXpert and Padchest evaluations.
>
> We indeed find that the average AUC improvement we see with our regularized model is consistent and statistically significant at a p=0.0001 level for both evaluations. Additionally, our model has superior performance on a majority of findings when compared head-to-head with the CheXzero and unregularized models, as seen by Tables 4,5,6, and Tables 9,10 in the appendix (pages 15, 18, and 19).
>
> > 2.2. Authors state that the proposed terms in equation 1) and 2) are weighted by two hyper parameters (λp and λt). Nevertheless, the code given is incorrect, as these hyperparameters are not present in the provided code.
>
> These two hyperparameters are included in the pseudocode (which has been moved to the appendix, Figure 6, page 13). These hyperparameters are referred to as “lambda_patch” for λp and “lambda_text” for λt. They are simply multiplied by their corresponding losses before being added to the CLIP loss to compute the total regularized loss.

---

> ### Author Response · Authors · 2023-02-07
> **Response to Reviewer bicQ (part 2)**
>
> > 3.1. The average improvement compared to the unregularized model is marginal (specially in Table 2).
>
> As we describe in our high-level summary, we re-ran our experiments with 1000 bootstraps using the single best checkpoint for each model to ensure that our AUC results were consistent on both the CheXpert and Padchest evaluations, and find that the average AUC improvement we see with our regularized model is consistent and statistically significant at a p=0.0001 level for both evaluations.See Tables 4,5,6, and Tables 9,10 in the appendix (pages 15, 18, and 19).
>
> > 3.2. The list of included methods in the experiments is insufficient...See for example [a-e].
>
> We add discussion of these approaches in a new “Related Works” section. While all of these methods describe efforts to promote more fine-grained alignment of images and text in CLIP-style models, we argue that our approach is sufficiently distinct and valuable:
>
> - References [c] and [d] rely on a separate region proposal/object detection network, so they are not directly comparable to our approach. These networks have not been equivalently validated in medical domains.
> - References [a], [b], and [e] do modify the contrastive loss as we do. However, the approaches in references [a] (GLoRIA)) and [e] (MGCA) do not directly encourage sparsity, unlike our method, potentially resulting in representations where all image tokens correspond to some degree to all text tokens and vice versa.
> - On the other hand, reference [b] (FILIP) does encourage sparsity. However, it does so more aggressibly and less flexibly than our method; by considering only the maximum similarity image patch for each text token and vice versa, the model may fail to capture relevant information for text tokens that span multiple image patches, or image patches that are relevant to many text tokens.
>
> We chose not to add these approaches to our evaluation because they are not trained on CXR data [b,c,d], shown to perform poorly in the CheXzero paper [a], or are not yet publicly available [e].
>
> > 3.3. Ablation study on the impact of the different values of the λp and λt hyperparameters should be included too.
>
> We include our results from a hyperparameter sweep over these hyperparameters in the appendix (table 3, page 15)
>
> > 3.4. I wonder why the patch size used in this work (Figures 3 and 4) is different from the size used in Boecking et al.
>
> We use the same method to produce patch embeddings as Boecking et al, and the resulting number of patches differs only because the image input sizes differ. In our preprocessing, images are resized to 224 by 224, whereas in Boecking et al. their input images were 512 by 512. We chose 224 by 224 because this is typical in many other implementations such as CLIP and CheXzero.
>
> **References**
> - [a] Huang SC, Shen L, Lungren MP, Yeung S. Gloria: A multimodal global-local representation learning framework for label-efficient medical image recognition. ICCV’21
> - [b] Yao L, Huang R, Hou L, Lu G, Niu M, Xu H, Liang X, Li Z, Jiang X, Xu C. FILIP: Fine-grained Interactive Language-Image Pre-Training. ICLR’21
> - [c] Li J, HE X, Wei L, Qian L, Zhu L, Xie L, Zhuang Y, Tian Q, Tang S. Fine-Grained Semantically Aligned Vision-Language Pre-Training. NeurIPS’22
> - [d] Zhong Y, Yang J, Zhang P, Li C, Codella N, Li LH, Zhou L, Dai X, Yuan L, Li Y, Gao J. Regionclip: Region-based language-image pretraining. CVPR’22
> - [e] Wang F, Zhou Y, Wang S, Vardhanabhuti V, Yu L. Multi-Granularity Cross-modal Alignment for Generalized Medical Visual

---

### Author Response · Authors · 2023-01-09
**General Response**

Thank you all for the valuable feedback on our submission. We have gotten a chance to review your comments, and over the coming two weeks, we will work to address the concerns you raised. In particular, we hope to be able to demonstrate the value and novelty of our method in context of other works, re-do our evaluations with multiple runs, and clarify several of the results that may have appeared unclear or unconvincing initially. We also aim to address each reviewer's comments point by point to ensure that all concerns have been addressed.

---

### Author Response · Authors · 2023-02-07
**Summary of response to reviewers**

We would like to thank the reviewers for the helpful and constructive feedback. We will provide a point-by-point response for each reviewer's comments, but first we would like to provide a high-level summary of our response. We would also like to thank the reviewers for their patience while we assembled this response, and look forward to continued discussion.

**Method novelty and motivation:** One point raised by the reviewers was with respect to the novelty and motivation of our proposed method, and we would like to provide some high-level comments on each:
- **Motivation:** The sparsity hypothesis behind our method is motivated by a domain-specific observation from the medical literature. When a radiologist reads and interprets these images they are, by definition, describing a specific and localized region of the image. For example, the finding of cardiomegaly (e.g. enlarged heart) is often described in an CXR report as “The cardiac silhouette is enlarged.”, which corresponds only to the lower left region of the chest.
- **Novelty:** While it is true that there are other methods that extend the base CLIP model to encourage alignment between image and text tokens, ours differs in several important ways. First, many of the alternative methods rely on external models (e.g. object detection networks) in order to perform the alignment, and thus are not directly comparable to the method proposed in our work. Furthermore, as we discuss in our detailed comments, many of the existing CLIP extensions that modify the contrastive loss (as we do), either do not aim to induce sparsity in the text-image alignments or do so in an overly aggressive manner (e.g. by allowing each text token to correspond to only one image token). In contrast, our method offers a flexible relaxation that allows one to dynamically set the extent of sparsity required using a set of hyperparameters determined by the dataset of interest.

**Relative performance improvement:** All reviewers commented on the modest improvement that our proposed regularization method offers. We understand this point and do not wish to oversell our results, however we have performed additional analysis requested by the reviewers to help better evaluate our contribution.
- **Bootstrapping:** We have performed a sensitivity analysis for the AUC metrics of our models and baselines, and have computed p-values via a bootstrap for all zero-shot classification comparisons in the manuscript.  We have added these results to the manuscript as seen in the appendix tables on pages 15, 18, and 19. On both evaluation datasets, we demonstrate that for the average AUC, and for the majority of findings, our regularized model improves performance at a significance threshold of p < 0.0001. We acknowledge the performance gains are modest (0.897 -> 0.903 for chexpert and 0.743 -> 0.755 for padchest), but we hope this additional analysis will give the reviewers confidence that the improvement is stable and robust. Additionally, even modest gains are important for high-stakes scenarios such as medicine.

- **Performance margin:** One fact that was not explicitly stated in the manuscript that we wish to highlight is that the models created in our paper, both the regularized and unregularized versions, represent a sizable advance in zero-shot CXR interpretation relative to the state of art in previous publications. For example even our unregularized model achieves an AUC of 0.907 vs. CheXzero’s AUC of 0.894 for Edema classification on the CheXpert evaluation (See tables 4,5, and 9 in the appendix for more examples of head-to-head comparisons against CheXzero).
- **Baseline comparison:** Even though our unregularized baseline architecture is based on Boecking et al., the version in our paper significantly outperforms the publicly available one from that manuscript. The off-the-shelf model from Boecking et al.  had the following results: Average AUC - 0.63631, Cardiomegaly - 0.63300, Edema - 0.58706, Consolidation - 0.70589, Atelectasis - 0.58698, Pleural Effusion - 0.66864. We have added these results to table 7 of the appendix, and explicitly mentioned this in the first paragraph of page 5 of our manuscript. We also describe the changes made to the Boecking et. al model beyond our regularization: we froze the first 8 layers of the text encoder before training further, altered the loss to remove their additional MLM loss, and trained on all lateral and frontal views of CXRs from the MIMIC-CXR dataset.

Taken as a whole, the models introduced in this work represent an advance on zero-shot interpretation of chest X-rays, which is an extremely important task in applied computer vision. Furthermore, we plan to make both the regularized and unregularized versions of our model publicly available for other researchers to use on this problem.

In light of our response, we ask the reviewers to reconsider their position if our claims are supported by the evidence we provide in the manuscript.

---

### Decision · Action_Editors · 2023-02-07

**Recommendation:** Reject

**Comment:**

This paper proposes a regularization scheme on the CLIP model. In particular, the authors propose to include two regularization terms on the text-patch embeddings. Results are reported on two classification benchmarks (CheXpert and PadChest).

By the end, this paper received 2 reject, and 1 leaning accepting recommendations. All the reviewers share some common concerns. 1) The novelty of the proposed method is limited. Many related works in the literature are not discussed. 2) The experimental results are weak, and the claimed improvements are very marginal. After checking the paper, the editor totally agrees with this. As the title suggests, this paper aims to propose a general technique that can improve CLIP-style models. However, the experiments are unconvincing and very small scale, and not evaluated and compared with other methods in a typical setting that CLIP-like models are evaluated. This paper needs a lot of efforts to make it ready for publication. Otherwise, the editor would suggest the authors to revise the title to reflect the fact that this paper is more medical-domain focused, and further improve the results to show the effectiveness of the method.

Overall, the editor would like to recommend rejection of the submission.

**Audience:**

The topic itself is interesting, as CLIP-style vision pre-training is a popular research topic. However, the empirical evaluation is not convincing.

**Claims And Evidence:**

The claims made in the submission are not well supported by the empirical evidence. In fact, all the reviewers agree that the experimental results are weak.